# Boosting Backdoor Attack with A Learnable Poisoning Sample Selection Strategy

## Abstract

Data-poisoning based backdoor attacks aim to inject backdoor into models by manipulating training datasets without controlling the training process of the target model. Existing backdoor attacks mainly focus on designing diverse triggers or fusion strategies to generate poisoned samples. However, all these attacks randomly select samples from the benign dataset to be poisoned, disregarding the varying importance of different samples. In order to select important samples to be poisoned from a global perspective, we first introduce a learnable poisoning mask into the regular backdoor training loss. Then we propose a **L**earnable **P**oisoning sample **S**election (LPS) strategy to learn the mask through a min-max optimization. During the two-player game, considering hard samples contribute more to the training process, the inner optimization maximizes loss $w.r.t.$ the mask to identify hard poisoned samples by impeding the training objective, while the outer optimization minimizes the loss $w.r.t.$ the model's weight to train the surrogate model. After several rounds of adversarial training, we finally select poisoned samples with high contribution. Extensive experiments on benchmark datasets demonstrate the effectiveness and efficiency of our LPS strategy in boosting the performance of various data-poisoning based backdoor attacks.

## 1 Introduction

Training deep neural networks (DNNs) necessitates a substantial volume of data. Considering the high cost of collecting or annotating massive data, users may resort to downloading publicly free datasets from open-sourced repositories or purchasing from third-party data suppliers. However, such unverified datasets may expose the DNNs to a serious threat of data-poisoning based backdoor attacks. By manipulating a subset of training samples, the adversary can inject a malicious backdoor into a trained model. During the inference stage, the backdoored model exhibits normal behavior when processing benign samples, but classifies any poisoned sample embedded with the trigger as the target class.

Several seminal backdoor attacks, including BadNets (Gu et al., 2019), Blended (Chen et al., 2017), SSBA (Li et al., 2021b), SIG (Barni et al., 2019), TrojanNN (Liu et al., 2018b), $etc.$, have demonstrated noteworthy success in compromising mainstream DNNs. Most of these attacks focus on developing diverse triggers ($e.g.$ patch triggers in BadNets, signal triggers in SIG) or employing fusion strategies of inserting the trigger into the benign samples ($e.g.$ alpha-blending in Blended, digital steganography in SSBA), which make the poisoned samples stealthy and effective. Nevertheless, all these attacks randomly select samples from the benign training dataset to create poisoned samples, disregarding the varying importance of different samples. Recent research (Koh & Liang, 2017; Katharopoulos & Fleuret, 2018; Paul et al., 2021) has substantiated that not all data contribute equally to the training of DNNs. Some specific samples carry greater significance for particular tasks. Consequently, several selection strategies, such as uncertainty-based (Coleman et al., 2020), influence function (Koh & Liang, 2017), forgetting events (Toneva et al., 2018), have been proposed to identify important samples in the context of various tasks, including coreset selection (Borsos et al., 2020; Killamsetty et al., 2021b;a), data valuation (Yoon et al., 2020; Nohyun et al., 2023; Just et al., 2023), and active learning (Chang et al., 2017; Kaushal et al., 2019).

It inspires us to investigate whether the performance of backdoor attacks could be boosted if the samples to be poisoned are selected according to well-designed strategies rather than random selection.

This underexplored issue has not received comprehensive attention within the backdoor learning community. Recently, Xia et al. (2022) proposes a filtering-and-updating (FUS) strategy based on the concept of forgetting events (Toneva et al., 2018). FUS filters out forgettable data from the poisoned samples, which are determined by forgetting events obtained through training on the entire poisoned dataset. And then these filtered data are replenished randomly from a candidate poisoned dataset. This process is iteratively employed to derive the final refined poisoned samples. However, it is noteworthy that FUS filters only in the poisoned subset at each step with a local perspective, thereby limiting its ability to comprehensively consider the entire poisoned dataset. Besides, the computation of forgetting events at each step requires the whole training process, resulting in a significant increase in computational cost. Therefore, *how to efficiently and effectively select samples for poisoning from the complete dataset with a global perspective, while maintaining generalization to various backdoor attacks* is still an urgent problem to be solved.

To address the aforementioned challenge, we propose a **L**earnable **P**oisoning sample **S**election strategy (LPS), which takes into account triggers, fusion strategies, and benign data simultaneously. The key intuition is that if a backdoor can be implanted into a model via **hard** poisoned samples during the training stage, the backdoor behavior can be effectively generalized to other **easy** poisoned samples at the inference stage. To achieve this objective, firstly, a learnable binary poisoning mask $m$ is introduced into the regular backdoor training loss (See Eq. (2)). Thereby finding hard samples can intuitively be obtained by impeding the backdoor training process (*i.e.*, maximize loss *w.r.t.* $m$), while the normal backdoor training can be achieved by minimizing loss *w.r.t.* $\theta$. To this end, we formulate the poisoning sample selection as a min-max optimization via an adversarial process. During the min-max two-player game, the inner maximization optimizes the mask to identify hard poisoned samples, while the outer minimization optimizes the model's weight to train a backdoored model based on the selected samples. By adversarially training the min-max problem over multiple rounds, we finally obtain the high-contributed poisoned samples that serve the malicious backdoor objective. The proposed LPS strategy can be naturally adopted in any off-the-shelf data-poisoning based backdoor attacks. Extensive evaluations with state-of-the-art backdoor attacks are conducted on benchmark datasets. The results demonstrate the superiority of our LPS strategy over both the random selection and the FUS strategy while resulting in significant computational savings.

The main contributions of this work are three-fold. **1)** We propose to identify hard poisoned samples from a global perspective. **2)** We propose a learnable poisoning sample selection strategy by formulating it as a min-max optimization problem. **3)** We provide extensive experiments to verify the effectiveness of the proposed selection strategy on significantly boosting existing data-poisoning backdoor attacks.

## 2 RELATED WORK

**Backdoor attack.** According to the threat model, existing backdoor attacks can be partitioned into two categories: *data-poisoning based* (Gu et al., 2019; Chen et al., 2017; Li et al., 2021b; Nguyen & Tran, 2021; Barni et al., 2019; Liu et al., 2018b) and *training-controllable based* (Nguyen & Tran, 2020; Doan et al., 2021a;b; Wang et al., 2022). In this work, we focus on the former threat model, where the adversary can only manipulate the training dataset and the training process is inaccessible. Thus, here we mainly review the related data-poisoning based attacks, and we refer readers to recent surveys (Wu et al., 2023; Li et al., 2020b; Wu et al., 2022b) for a detailed introduction to training-controllable based backdoor attacks. BadNets (Gu et al., 2019) was the first attempt to stamp a patch on the benign image as the poisoned image, revealing the existence of backdoor in deep learning. Blended (Chen et al., 2017) used the alpha blending strategy to make the trigger invisible to evade human inspection. SIG (Barni et al., 2019) generated a ramp or triangle signal as the trigger. TrojanNN attack (Liu et al., 2018b) optimized the trigger by maximizing its activation on selected neurons related. SSBA (Li et al., 2021b) adopted a digital stenography to fuse a specific string into images by autoencoder, to generate sample-specific triggers. Subsequently, more stealthy and effective attacks (Zeng et al., 2021b; Zhang et al., 2022; Salem et al., 2022; Turner et al., 2019; Souri et al., 2022; Nguyen & Tran, 2020; Doan et al., 2022) have been successively proposed. Meanwhile, some defence methods (Tran et al., 2018; Huang et al., 2022; Wang et al., 2023; Chai & Chen, 2022; Chen et al., 2019; Doan et al., 2023; Zeng et al., 2022) have been proposed as shields to resist attacks. The commonality of the above attacks is that they focus on designing triggers or the fusion strategy and simply adopt the random selection strategy while overlooking how to select benign samples for

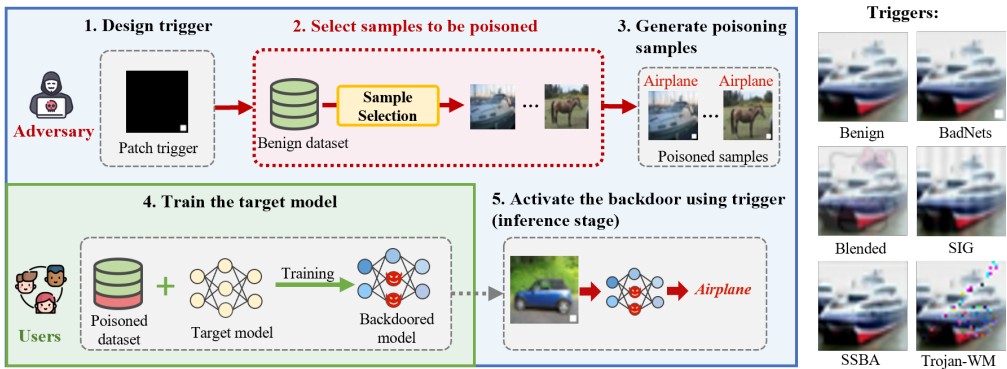

Figure 1: The general procedure of data-poisoning based backdoor attack and examples of representative triggers.

generating poisoned samples. Instead, we aim to boost existing data-poisoning backdoor attacks through a learnable poisoning sample selection strategy depending on the trigger and benign data.

**Poisoning sample selection in backdoor attack.** Poisoning sample selection has not been comprehensively studied in the backdoor attack community. Xia et al. (2022) proposes a filtering-and-updating strategy, which iteratively filters and updates selected samples from a local perspective. FUS filters out easily forgotten samples from the poisoned subset based on forgetting events (Toneva et al., 2018), which are obtained by completely training DNNs on the entire poisoned dataset. And then additional poisoned samples are randomly sampled from a candidate poisoned set to update the pool. This process are iteratively employed to obtain the final refined poisoned samples. As a pioneering work, FUS improves the performance of backdoor attacks compared to random selection strategy. However, FUS overlooks considering the entire poisoned dataset from a global perspective and the computation of forgetting events at each step requires tens of times more computing cost.

## 3 PRELIMINARY

**Threat model.** Following existing attacks (Doan et al., 2021b; Wang et al., 2022), we assume that the adversary has full control of the dataset, which is a widely adopted threat model in data-poisoning based backdoor attacks (Li et al., 2021b; Souri et al., 2022). In this scenario, the adversary, as the data provider, has access to select samples from the entire dataset for poisoning. It is noteworthy that the user, who downloads the dataset, retains control over the training process, whereas the adversary has no access to other training components, such as model architecture or hyperparameters of the target model. Therefore, the adversary resorts to a surrogate model to simulate the backdoor attack and select poisoned samples, which is the same settings adopted in Xia et al. (2022).

**General procedure of data-poisoning based backdoor attacks.** Here we define the general procedure of data-poisoning based backdoor attacks. As shown in Fig. 1, it consists of five steps:

❶ **Design trigger (by adversary).** The first step of backdoor attack is to design a trigger $\epsilon$, of which the format could be diverse in different applications, such as a patch (Gu et al., 2019) or a specific image (Chen et al., 2017), as shown in the right part of Fig. 1.

❷ **Select samples to be poisoned (by adversary).** Let $\mathcal{D} = \{(\boldsymbol{x}_i, y_i)\}_{i=1}^{|\mathcal{D}|}$ denote the original benign training dataset that contains $|\mathcal{D}|$ i.i.d. samples, where $\boldsymbol{x}_i \in \mathcal{X}$ denotes the input image, $y_i \in \mathcal{Y} = \{1, \ldots, K\}$ is the ground-truth label of $\boldsymbol{x}_i$. There are $K$ candidate classes, and the size of class $k$ is denoted as $n_k$. For clarity, we assume that all training samples are ranked following the class indices, i.e., (samples of class 1), (samples of class 2), . . . , (samples of class $K$). To ensure stealthiness and avoid harm to clean accuracy, the adversary often selects a small fraction of benign samples to be poisoned. Here we define a binary vector $\boldsymbol{m} = [m_1, m_2, \ldots, m_{|\mathcal{D}|}] \in \{0, 1\}^{|\mathcal{D}|}$ to represent the poisoning mask, where $m_i = 1$ indicates that $\boldsymbol{x}_i$ is selected to be poisoned and $m_i = 0$ means not selected. We denote $\alpha := \sum_{i=1}^{|\mathcal{D}|} m_i / |\mathcal{D}|$ as the poisoning ratio.

❸ **Generate poisoned samples (by adversary).** Given the trigger $\epsilon$ and the selected sample $\boldsymbol{x}_i$ (i.e., $m_i = 1$), the adversary will design some strategies to fuse $\epsilon$ into $\boldsymbol{x}_i$ to generate the poisoned sample $\tilde{\boldsymbol{x}}_i = g(\boldsymbol{x}_i, \epsilon)$, with $g(\cdot, \cdot)$ denoting the fusion operator (e.g. the alpha-blending used in

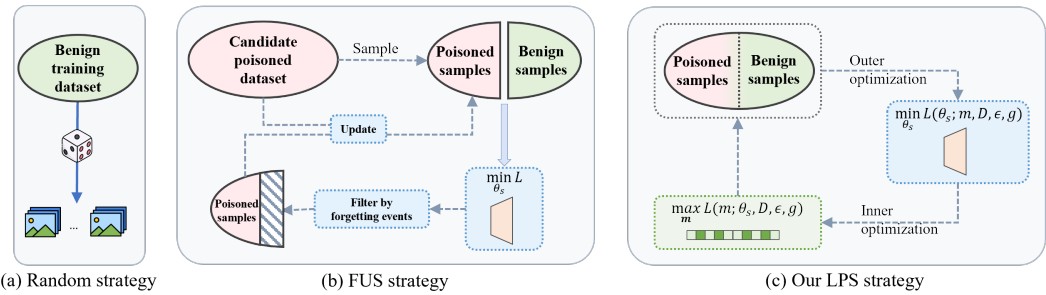

Figure 2: Different poisoning sample selection strategies.

Blended (Chen et al., 2017)). Besides, the adversary has the authority to change the original ground-truth label $y_i$ to the target label $\tilde{y}_i$. If target labels remain the same for all poisoned samples (*i.e.*, $\tilde{y}_i = y_t$), it is called *all-to-one* attack. If target labels have differnt types (*e.g.*, $\tilde{y}_i = y_i + 1$), it is called *all-to-all* attack. If adversary does not change the ground-truth label (*i.e.*, $\tilde{y}_i = y_i$), it is called *clean label* attack. Thus, the generated poisoned training dataset could be denoted as $\tilde{\mathcal{D}} = \{(\boldsymbol{x}_i, y_i)|_{m_i=0}\} \cup \{(\tilde{\boldsymbol{x}}_i, \tilde{y}_i)|_{m_i=1}\}$.

❹ **Train the target model (by user).** Given the poisoned training dataset $\tilde{\mathcal{D}}$, the user trains the target model $f_{\boldsymbol{\theta}_t}$ by minimizing the following loss function:

$$\mathcal{L}(\boldsymbol{\theta}_t; \tilde{\mathcal{D}}) = \frac{1}{|\tilde{\mathcal{D}}|} \sum_{(\boldsymbol{x},y)\in\tilde{\mathcal{D}}} \ell(f_{\boldsymbol{\theta}_t}(\boldsymbol{x}), y)) \tag{1}$$

$$\equiv \mathcal{L}(\boldsymbol{\theta}_t; \mathcal{D}, \boldsymbol{m}, \boldsymbol{\epsilon}, g) = \frac{1}{|\mathcal{D}|} \sum_{i=1}^{|\mathcal{D}|} \Big[ (1 - m_i) \cdot \ell(f_{\boldsymbol{\theta}_t}(\boldsymbol{x}_i), y_i)) + m_i \cdot \ell(f_{\boldsymbol{\theta}_t}(\tilde{\boldsymbol{x}}_i), y_t) \Big], \tag{2}$$

where $\ell(\cdot, \cdot)$ is the loss function for an individual sample, such as cross-entropy loss. In Eq. (2), we extend Eq. (1) by introducing binary poisoning mask $\boldsymbol{m}$ that is described in step 2.

❺ **Activate the backdoor using the trigger during the inference stage (by the adversary)** Given the trained model $f_{\boldsymbol{\theta}_t}$, the adversary expects to activate the injected backdoor using the trigger $\boldsymbol{\epsilon}$, *i.e.*, fooling $f_{\boldsymbol{\theta}_t}$ to predict any poisoned sample $g(\boldsymbol{x}_i, \boldsymbol{\epsilon})$ as the target label $\tilde{y}_i$.

Most backdoor attacks concentrate on designing diverse triggers (*i.e.*, step 1) or fusion strategies (*i.e.*, step 3). These attacks typically randomly select samples for poisoning (*i.e.*, step 2, shown in Fig. 2a), neglecting the unequal influence of each poisoned sample for backdoor injection. The recent FUS strategy (Xia et al., 2022), illustrated in Fig. 2b, involves filtering out less significant poisoning samples based on forgetting events (Toneva et al., 2018) and updating the poisoned subset by resampling from the candidate. However, it operates from a local perspective rather than considering all alternate poisoned samples together. Therefore, the development of a poisoning sample selection strategy encompassing the entire dataset remains a challenge.

## 4 METHODOLOGY: LEARNABLE POISONING SAMPLE SELECTION STRATEGY

This work aims to design a poisoning sample selection strategy to boost the performance of backdoor attacks. As the target model $f_{\boldsymbol{\theta}_t}$ is agnostic to the adversary, we resort to a surrogate model $f_{\boldsymbol{\theta}_s}$ as an alternative. In order to select samples for poisoning from the entire dataset with a global perspective, we directly generate the poisoning mask $\boldsymbol{m}$ in step 2. We suppose that if a backdoor can be implanted into the model through *hard* poisoned samples, the backdoor can be generalized to other more *easy* poisoned samples during the inference stage. To find such hard poisoned samples, an intuitive way is to hinder the normal backdoor training, *i.e.*, maximizing the loss in Eq. (2) *w.r.t.* $\boldsymbol{m}$. To combine it with the normal training objective (*i.e.*, minimizing Eq. (2) *w.r.t.* $\boldsymbol{\theta}_s$), we propose a **L**earnable **P**oisoning sample **S**election (LPS) strategy to learn the poisoning mask $\boldsymbol{m}$ along with the surrogate model's parameters $\boldsymbol{\theta}_s$ through a min-max optimization, as shown in Fig. 2c:

$$\min_{\boldsymbol{\theta}_s} \max_{\boldsymbol{m}\in\{0,1\}^{|\mathcal{D}|}} \Big\{ \mathcal{L}(\boldsymbol{\theta}_s, \boldsymbol{m}; \mathcal{D}, \boldsymbol{\epsilon}, g) \quad \text{s.t. } \boldsymbol{H}\boldsymbol{m} = \tilde{\alpha} \cdot \boldsymbol{\mu} \Big\}, \tag{3}$$

where $\mathcal{L}$ is extended loss including poisoning mask that defined in Eq. (2). $\boldsymbol{H} \in \{0,1\}^{K \times |\mathcal{D}|}$ is defined as: in the $k$-th row, the entries $\boldsymbol{H}(k, \sum_{j=1}^{k-1} n_j + 1 : \sum_{j=1}^{k} n_j) = 1$, while other entries are 0. $\tilde{\alpha} = \frac{\alpha \cdot |\mathcal{D}|}{\sum_{k \neq y_t} n_k}$ and $\tilde{\alpha} n_k$ is integer for all $k$. $\boldsymbol{\mu} = [\mu_1; \mu_2; \dots; \mu_K] \in \mathbb{N}^K$ is defined as: if $k \neq y_t$, then $\mu_k = n_k$, otherwise $\mu_k = 0$. This equation captures three constraints, including: **1)** $\alpha \cdot |\mathcal{D}|$ samples are selected to be poisoned; **2)** the target class samples cannot be selected to be poisoned; **3)** each non-target class has the same selected ratio $\tilde{\alpha}$ to encourage the diversity of selected samples. Note that here we only consider the setting of *all-to-one* attack, but the constraint can be flexibly adjusted for *all-to-all* and *clean label* settings.

**Remark.** This min-max objective function (3) is designed for finding hard poisoned samples with high-contribution for backdoor injection via an adversarial process. Specifically, the inner loop encourages to select hard poisoned samples for the given model's parameters $\boldsymbol{\theta}_s$ by maximizing the loss $w.r.t.$ $\boldsymbol{m}$, while the outer loop aims to update $\boldsymbol{\theta}_s$ by minimizing the loss $w.r.t.$ $f_{\boldsymbol{\theta}_s}$ to ensure that a good backdoored model can be still learned, even based on the hard poisoning mask $\boldsymbol{m}$. Thus, the two-player game between $\boldsymbol{m}$ and $\boldsymbol{\theta}_s$ is expected to encourage the selected samples to bring in good backdoor effect.

---

**Algorithm 1** LPS strategy via min-max optimization

**Input:** Benign training dataset $\mathcal{D}$, architecture of the surrogate model $f_{\boldsymbol{\theta}_s}$, maximal iterations $T$, poisoning ratio $\alpha$, trigger $\boldsymbol{\epsilon}$, fusion operator $g$

**Output:** poisoning mask $\boldsymbol{m}$
1: Randomly initialize $\boldsymbol{m}_s^{(0)}$, $\boldsymbol{\theta}_s^{(0)}$
2: **for each** iteration $t = 0$ to $T - 1$ **do**
3:     ▷ Given $\boldsymbol{m}^{(t)}$, update $\boldsymbol{\theta}_s^{(t+1)}$ by solving outer sub-problem in Eq. (4).
4:     ▷ Given $\boldsymbol{\theta}_s^{(t+1)}$, update $\boldsymbol{m}^{(t+1)}$ by solving inner sub-problem in Eq. (5).
5: **end for**
6: **return** $\boldsymbol{m}_T$

---

**Optimization.** As summarized in Algorithm 1, the min-max optimization (3) could be efficiently solved by alternatively updating $\boldsymbol{m}$ and $\boldsymbol{\theta}_s$ as follows:

♦ **Outer minimization**: given $\boldsymbol{m}$, $\boldsymbol{\theta}_s$ could be updated by solving the following sub-problem:

$$\boldsymbol{\theta}_s \in \arg\min_{\boldsymbol{\theta}_s} \mathcal{L}(\boldsymbol{\theta}_s; \boldsymbol{m}, \mathcal{D}, \boldsymbol{\epsilon}, g). \tag{4}$$

It could be optimized by the standard back-propagation method with stochastic gradient descent (SGD) (Bottou & Bousquet, 2007). Here we update $\boldsymbol{\theta}_s$ for one epoch in each iteration.

♦ **Inner maximization**: given $\boldsymbol{\theta}_s$, $\boldsymbol{m}$ could be achieved by solving the maximization problem as:

$$\boldsymbol{m} \in \arg\max_{\boldsymbol{m} \in \{0,1\}^{|\mathcal{D}|}} \left\{ \mathcal{L}(\boldsymbol{m}; \boldsymbol{\theta}_s, \mathcal{D}, \boldsymbol{\epsilon}, g), \text{ s.t. } \boldsymbol{H}\boldsymbol{m} = \tilde{\alpha} \cdot \boldsymbol{\mu} \right\}. \tag{5}$$

Although it is a constrained binary optimization problem, it is easy to decompose it into $K$ independent linear optimization sub-problems, $i.e.$, for $\forall k \in \{1, 2, \dots, K\}$,

$$\max_{\boldsymbol{m}^k \in \{0,1\}^{n_k}} \frac{1}{|\mathcal{D}|} \sum_{i=1}^{n_k} m_i^k \cdot \left[ \ell(f_{\boldsymbol{\theta}_s}(\tilde{\boldsymbol{x}}_{ik}), y_t) - \ell(f_{\boldsymbol{\theta}_s}(\boldsymbol{x}_{ik}), y_{ik}) \right], \text{ s.t. } \boldsymbol{1}_{n_k}^\top \boldsymbol{m}^k = \tilde{\alpha} \cdot \mu_k, \tag{6}$$

where $\boldsymbol{m}^k$ denotes the sub-mask vector of $\boldsymbol{m}$ corresponding to samples of class $k$ and $(\boldsymbol{x}_{ik}, y_{ik})$ is the $i$-th sample of the set $\mathcal{D}_k = \{(x, y) \mid y = k, (x, y) \in \mathcal{D}\}$ that is the subset with class $k$. Obtaining the optimal solution is a straightforward process, which involves computing $\delta_i = \ell(f_{\boldsymbol{\theta}_s}(\tilde{\boldsymbol{x}}_i), y_t) - \ell(f_{\boldsymbol{\theta}_s}(\boldsymbol{x}_i), y_i)$ for all samples, then for each class k sorting the subset $\Delta_k = \{\delta_i \mid (x_i, y_i) \in \mathcal{D}_k\}$ in descending order and selecting the corresponding top-$(\tilde{\alpha} \cdot \mu_k)$ samples. The decomposition of Eq. (5) and the solution of Eq. (6) are proved in Appendix B.

**Remark.** Above selection criterion ensures that the selected poisoned samples exhibit a large loss gap between $\ell(f_{\boldsymbol{\theta}_s}(\tilde{\boldsymbol{x}}_i), y_t)$ and $\ell(f_{\boldsymbol{\theta}_s}(\boldsymbol{x}_i), y_i)$ in each iteration. Such poisoned samples with large backdoor loss $\ell(f_{\boldsymbol{\theta}_s}(\tilde{\boldsymbol{x}}_i), y_t)$ are characterized by being difficult to learn, which aligns with the conclusion in Coleman et al. (2020); Nguyen et al. (2022) that hard-to-learn samples contribute more to the learning process. Meanwhile, the clean loss $\ell(f_{\boldsymbol{\theta}_s}(\boldsymbol{x}_i), y_i)$ of corresponding benign samples are small, indicating that they are relatively easy to learn, which demonstrates that the original benign feature of poisoned samples should be more conspicuous to enhance the generalization of backdoor to other samples. A more detailed analysis can be found in Section 6.

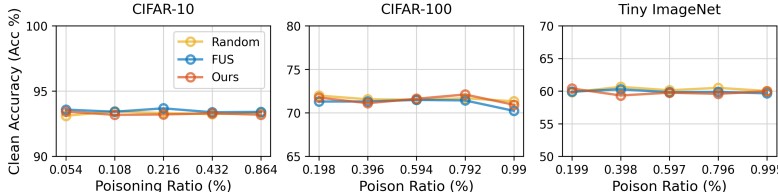

Figure 3: Clean accuracy of Blended attack with different backdoor sample selection strategies.

## 5 EXPERIMENTS

### 5.1 EXPERIMENTAL SETTINGS

**Implementation details.** For the inner minimization of the surrogate model and the train of target model, we adopt SGD optimizer with weight decay set as $5e-4$, the batch size set as 128, the initial learning rate set as $0.01$ and reduced by 10 after 35 and 55 epochs, respectively. The maximal iteration $T$ is set as 15. The training epoch for the target model is set as 100.

**Datasets and models.** We evaluate on three commonly used benchmark datasets: CIFAR-10, CIFAR-100 (Krizhevsky et al., 2009) and Tiny-ImageNet (Le & Yang, 2015). The surrogate model and target model are chosen from ResNet (He et al., 2016), VGG (Simonyan & Zisserman, 2015), MobileNet (Howard et al., 2017) and DenseNet Huang et al. (2017).

**Baselines of poisoning sample selection.** We compare our proposed LPS strategy with two existing poisoning sample selection strategies: *Random* and *Filtering-and-Updating Stragtegy (FUS)*[1] (Xia et al., 2022). Following the original setting in FUS, we set 10 overall iterations and 60 epochs for updating the surrogate model in each iteration.

**Backdoor attacks.** We consider five representative backdoor attacks: 1) visible triggers: BadNets (Gu et al., 2019), Blended (Chen et al., 2017); SIG (Barni et al., 2019); 2) optimized triggers: Trojan-Watermark (Trojan-WM) (Liu et al., 2018b); 3) sample-specific triggers: SSBA (Li et al., 2021b). In addition, we consider 3 poisoning label types: all-to-one, all-to-all and clean label. We visualize different triggers with the same benign image in Fig. 1. The detailed settings of each attack can been found in **Appendix** E.

**Backdoor defenses.** We select six representative backdoor defenses to evaluate the resistance of above attack methods with different poisoning sample selection strategies, including Fine-Tuning (FT), Fine-Pruning (FP) (Liu et al., 2018a), Anti-Backdoor Learning (ABL) (Li et al., 2021a), Channel Lipschitzness Pruning (CLP) (Zheng et al., 2022), Neural Attention Distillation (NAD) (Li et al., 2020a), Implicit Backdoor Adversarial Unlearning (I-BAU) (Zeng et al., 2021a). The detailed settings of each defense can be found in **Appendix** E.

### 5.2 MAIN RESULTS

We evaluate our LPS strategy under various experimental settings, including comparisons with baseline strategies on various attacks and poisoning ratios, comparisons on different datasets and resistance to defenses. Besides, we find that due to the low poisoning ratios, the impacts of different poisoning sample selection strategies on the clean accuracy are almost similar (as shown in Fig. 3). Thus, for clarity, we omit ACC in most result tables, except for Tab. 4. Three random trials are conducted for the main experiments to report the mean and standard deviation. More results about different models can be found in **Appendix** F.

**Compare with state-of-the-art baselines.** To verify the effectiveness of our proposed LPS strategy, we first compare with two existing strategies on CIFAR-10, in which the surrogate model is ResNet-18 and the target model is ResNet-34. Different from Xia et al. (2022), we conduct experiments under low poisoning ratios ($< 1\%$), which is more stealthy and more likely to escape human inspection. The attack success rate is shown in Tab. 1, where *#Img/Cls* denotes the number of samples to be poisoned

---

[1]Note that in the experiments reported in Xia et al. (2022), FUS appended the generated poisoned samples onto the original benign dataset, rather than replacing the selected benign samples, leading to $|\tilde{\mathcal{D}}| \geq |\mathcal{D}|$. To ensure fair comparison, we change it to the traditional setting in existing attacks that the selected benign samples to be poisoned are replaced by the generated samples, thus $|\tilde{\mathcal{D}}| = |\mathcal{D}|$.

Table 1: Attack success rate (%) on CIFAR-10, where the surrogate and target model are ResNet-18 and ResNet-34 respectively. **Bold** means the best.

| | Dataset: CIFAR-10 | Surrogate: ResNet-18 $\Longrightarrow$ Target: ResNet-34 | | | |
|---|---|---|---|---|---|
| Attack | Pratio (#Img/Cls) | 0.054% (#3) | 0.108% (#6) | 0.216% (#12) | 0.432% (#24) | 0.864% (#48) |
| BadNets (all-to-one) | Random | **0.86** $\pm$ 0.09 | 1.71 $\pm$ 0.48 | 62.57 $\pm$ 5.15 | 81.71 $\pm$ 1.51 | 89.21 $\pm$ 1.05 |
| | FUS | 0.75 $\pm$ 0.08 | 1.37 $\pm$ 0.22 | 64.67 $\pm$ 5.88 | 83.41 $\pm$ 2.09 | 90.05 $\pm$ 0.34 |
| | LPS (Ours) | 0.77 $\pm$ 0.04 | **5.70** $\pm$ 1.77 | **76.41** $\pm$ 5.03 | **85.77** $\pm$ 5.43 | **91.62** $\pm$ 1.25 |
| BadNets (all-to-all) | Random | 0.69 $\pm$ 0.05 | 0.73 $\pm$ 0.06 | 0.93 $\pm$ 0.22 | 39.91 $\pm$ 14.35 | 75.54 $\pm$ 2.84 |
| | FUS | **0.72** $\pm$ 0.01 | 0.75 $\pm$ 0.02 | 1.03 $\pm$ 0.13 | 33.37 $\pm$ 2.60 | 76.76 $\pm$ 0.24 |
| | LPS (Ours) | 0.70 $\pm$ 0.10 | **0.76** $\pm$ 0.05 | **36.64** $\pm$ 2.71 | **66.95** $\pm$ 1.02 | **80.18** $\pm$ 2.08 |
| Blended (all-to-one) | Random | 8.87 $\pm$ 2.75 | 23.69 $\pm$ 3.09 | 50.65 $\pm$ 4.05 | 75.67 $\pm$ 1.51 | 89.47 $\pm$ 0.93 |
| | FUS | **10.51** $\pm$ 2.01 | 22.29 $\pm$ 0.81 | 51.13 $\pm$ 2.83 | 80.46 $\pm$ 1.18 | 92.11 $\pm$ 0.79 |
| | LPS (Ours) | 9.09 $\pm$ 3.54 | **29.84** $\pm$ 3.86 | **64.6** $\pm$ 4.51 | **87.16** $\pm$ 0.84 | **97.53** $\pm$ 0.19 |
| Blended (all-to-all) | Random | 2.48 $\pm$ 0.11 | 4.05 $\pm$ 0.66 | 9.32 $\pm$ 1.25 | 37.33 $\pm$ 5.01 | 67.54 $\pm$ 1.12 |
| | FUS | 2.40 $\pm$ 0.16 | 4.17 $\pm$ 0.60 | 6.67 $\pm$ 0.49 | 29.54 $\pm$ 2.34 | 64.90 $\pm$ 2.02 |
| | LPS (Ours) | **3.35** $\pm$ 0.45 | **7.37** $\pm$ 0.78 | **34.6** $\pm$ 2.34 | **60.12** $\pm$ 1.60 | **72.92** $\pm$ 1.00 |
| SIG (clean label) | Random | 3.48 $\pm$ 0.74 | 6.16 $\pm$ 1.74 | 11.98 $\pm$ 0.75 | 18.72 $\pm$ 3.18 | 36.46 $\pm$ 5.34 |
| | FUS | 3.30 $\pm$ 0.59 | 8.67 $\pm$ 2.10 | 16.06 $\pm$ 3.16 | 28.50 $\pm$ 1.14 | 46.99 $\pm$ 8.77 |
| | LPS (Ours) | **11.38** $\pm$ 1.50 | **19.00** $\pm$ 1.66 | **32.67** $\pm$ 3.06 | **51.32** $\pm$ 4.17 | **65.77** $\pm$ 5.80 |
| SSBA (all-to-one) | Random | 1.01 $\pm$ 0.12 | **1.05** $\pm$ 0.04 | 2.06 $\pm$ 0.15 | 20.34 $\pm$ 5.58 | 60.36 $\pm$ 2.42 |
| | FUS | **1.10** $\pm$ 0.16 | 1.04 $\pm$ 0.28 | 2.02 $\pm$ 0.45 | 16.81 $\pm$ 3.47 | 60.64 $\pm$ 3.29 |
| | LPS (Ours) | 0.98 $\pm$ 0.17 | 1.03 $\pm$ 0.05 | **2.30** $\pm$ 0.51 | **22.92** $\pm$ 2.74 | **64.39** $\pm$ 2.96 |
| Trojan-WM (all-to-one) | Random | 3.39 $\pm$ 1.37 | 23.26 $\pm$ 11.74 | 80.04 $\pm$ 7.19 | 94.96 $\pm$ 1.92 | 98.27 $\pm$ 0.34 |
| | FUS | 3.07 $\pm$ 1.62 | 19.22 $\pm$ 6.12 | 78.85 $\pm$ 4.70 | 96.59 $\pm$ 1.57 | 99.25 $\pm$ 0.38 |
| | LPS (Ours) | **3.66** $\pm$ 0.33 | **33.77** $\pm$ 10.47 | **94.32** $\pm$ 0.81 | **99.77** $\pm$ 0.06 | **99.97** $\pm$ 0.01 |

Table 2: Attack success rate (%) on CIFAR-100, where the surrogate and target model are ResNet-18 and ResNet-34 respectively. **Bold** means the best.

| | Dataset: CIFAR-100 | Surrogate: ResNet-18 $\Longrightarrow$ Target: ResNet-34 | | | |
|---|---|---|---|---|---|
| Attack | Pratio (#Img/Cls) | 0.198% (#1) | 0.396% (#2) | 0.594% (#3) | 0.792% (#4) | 0.99 % (#5) |
| BadNets (all-to-one) | Random | 8.09 $\pm$ 2.31 | 36.74 $\pm$ 6.22 | 50.68 $\pm$ 2.68 | 59.50 $\pm$ 4.56 | 64.81 $\pm$ 5.97 |
| | FUS | 10.41 $\pm$ 4.20 | 43.60 $\pm$ 6.79 | 51.06 $\pm$ 6.74 | 62.28 $\pm$ 6.22 | 68.34 $\pm$ 6.30 |
| | LPS (Ours) | **17.98** $\pm$ 2.58 | **52.02** $\pm$ 4.05 | **58.46** $\pm$ 1.87 | **63.49** $\pm$ 5.90 | **70.45** $\pm$ 3.49 |
| Blended (all-to-one) | Random | 37.53 $\pm$ 2.23 | 59.98 $\pm$ 1.09 | 68.53 $\pm$ 1.26 | 77.37 $\pm$ 0.67 | 81.47 $\pm$ 0.26 |
| | FUS | **38.65** $\pm$ 1.58 | 65.75 $\pm$ 0.98 | 69.04 $\pm$ 5.40 | 82.25 $\pm$ 0.69 | 86.14 $\pm$ 0.46 |
| | LPS (Ours) | 38.64 $\pm$ 1.72 | **66.94** $\pm$ 1.75 | **81.73** $\pm$ 1.73 | **89.88** $\pm$ 1.19 | **93.29** $\pm$ 0.67 |
| SIG (clean label) | Random | 2.79 $\pm$ 0.44 | 6.09 $\pm$ 0.99 | 14.3 $\pm$ 2.38 | 22.08 $\pm$ 3.28 | 43.95 $\pm$ 1.35 |
| | FUS | 3.79 $\pm$ 1.12 | **7.80** $\pm$ 1.60 | 15.84 $\pm$ 2.50 | N/A | N/A |
| | LPS (Ours) | **4.49** $\pm$ 1.43 | 7.01 $\pm$ 1.73 | **16.11** $\pm$ 1.99 | **25.12** $\pm$ 1.37 | **46.43** $\pm$ 0.55 |
| SSBA (all-to-one) | Random | 1.42 $\pm$ 0.24 | 7.45 $\pm$ 1.62 | 18.73 $\pm$ 3.33 | 31.61 $\pm$ 0.63 | 43.37 $\pm$ 1.77 |
| | FUS | **1.51** $\pm$ 0.40 | 7.99 $\pm$ 1.11 | 18.44 $\pm$ 1.51 | 33.35 $\pm$ 1.06 | 44.00 $\pm$ 2.66 |
| | LPS (Ours) | 1.49 $\pm$ 0.10 | **8.03** $\pm$ 1.09 | **21.46** $\pm$ 1.81 | **34.12** $\pm$ 2.85 | **48.77** $\pm$ 3.18 |
| Trojan-WM (all-to-one) | Random | 39.44 $\pm$ 4.24 | 68.64 $\pm$ 1.83 | 82.13 $\pm$ 0.47 | 88.08 $\pm$ 0.93 | 91.16 $\pm$ 1.52 |
| | FUS | 39.74 $\pm$ 2.42 | 75.43 $\pm$ 3.23 | 84.80 $\pm$ 0.79 | 92.58 $\pm$ 0.95 | 93.87 $\pm$ 0.33 |
| | LPS (Ours) | **44.90** $\pm$ 3.51 | **84.75** $\pm$ 3.24 | **96.36** $\pm$ 1.18 | **98.16** $\pm$ 0.33 | **99.30** $\pm$ 0.16 |

per class for all-to-one setting, and *pratio* is short for poisoning ratio. **1) From a global view**, we observe that LPS strategy outperforms the baselines under most of the settings. For example, with $0.216\%$ poisoning ratio, LPS strategy can boost BadNets (all-to-all) by $30.61\%$ compared to FUS, and Blended (all-to-one) can be improved by $13.53\%$. **2) From the perspective of poisoning ratios**, LPS strategy can be widely applied to different poisoning ratios, but the degree of improvement is also related to the poisoning ratio. Specifically, when the poisoning ratio is extremely low (*e.g.*, 1 Img/Cls, $0.054\%$ pratio), although the improvement of our method is not obvious compared with other strategies due to the attack itself being weak, it also shows similar results. However, once the poisoning ratio is increased, LPS shows a strong advantage over other strategies. **3) From the perspective of attacks**, our LPS strategy consistently improves different types of triggers and poisoning labels, demonstrating that LPS strategy is widely applicable to various backdoor attacks.

**Comparisons across different datasets.** To verify whether our proposed LPS strategy supports larger datasets (more images and classes, larger image size), we also evaluate these three strategies on CIFAR-100 and Tiny-ImageNet. The results in Tabs. 2 and 3 further demonstrate the superiority of LPS strategy to both the random selection and the FUS strategy.

Table 3: Attack success rate (%) on Tiny-ImageNet, where the surrogate and target model are ResNet-18 and ResNet-34 respectively. **Bold** means the best.

| | Dataset: Tiny-ImageNet | Surrogate: ResNet-18 $\Longrightarrow$ Target: ResNet-34 | | | |
|---|---|---|---|---|---|
| Attack | Pratio (#Img/Cls) | 0.199% (#1) | 0.398% (#2) | 0.597% (#3) | 0.796% (#4) | 0.995% (#5) |
| BadNets (all-to-one) | Random | 4.93 $\pm$ 6.19 | 37.18 $\pm$ 6.61 | 42.98 $\pm$ 1.89 | 48.91 $\pm$ 3.46 | 60.52 $\pm$ 2.35 |
| | FUS | **5.44** $\pm$ 3.54 | 32.93 $\pm$ 1.69 | 43.74 $\pm$ 3.67 | 48.72 $\pm$ 3.58 | 60.76 $\pm$ 4.72 |
| | LPS (Ours) | 5.21 $\pm$ 3.10 | **38.05** $\pm$ 2.26 | **47.21** $\pm$ 3.90 | **49.34** $\pm$ 3.41 | **61.22** $\pm$ 2.12 |
| Blended (all-to-one) | Random | 66.73 $\pm$ 0.52 | 78.79 $\pm$ 0.63 | 84.87 $\pm$ 1.50 | 87.81 $\pm$ 0.72 | 89.96 $\pm$ 0.43 |
| | FUS | 70.95 $\pm$ 1.47 | 82.01 $\pm$ 0.50 | 88.38 $\pm$ 0.94 | 90.70 $\pm$ 1.37 | 93.19 $\pm$ 0.39 |
| | LPS (Ours) | **82.76** $\pm$ 2.52 | **93.55** $\pm$ 0.45 | **96.20** $\pm$ 0.11 | **97.65** $\pm$ 0.10 | **98.08** $\pm$ 0.09 |
| SIG (all-to-one) | Random | 61.80 $\pm$ 3.30 | 81.15 $\pm$ 0.62 | 87.87 $\pm$ 1.83 | 90.80 $\pm$ 0.55 | 92.77 $\pm$ 0.95 |
| | FUS | 60.02 $\pm$ 1.76 | 84.95 $\pm$ 2.53 | 86.36 $\pm$ 6.11 | 92.47 $\pm$ 1.41 | 94.56 $\pm$ 0.59 |
| | LPS (Ours) | **62.90** $\pm$ 3.07 | **91.57** $\pm$ 2.00 | **96.59** $\pm$ 1.07 | **98.02** $\pm$ 0.45 | **98.97** $\pm$ 0.17 |
| SSBA (all-to-one) | Random | 34.34 $\pm$ 2.93 | 60.05 $\pm$ 3.32 | 76.09 $\pm$ 0.88 | 81.60 $\pm$ 0.25 | 85.65 $\pm$ 0.30 |
| | FUS | **34.80** $\pm$ 0.71 | 60.68 $\pm$ 1.58 | 76.83 $\pm$ 1.55 | 84.53 $\pm$ 0.54 | 88.48 $\pm$ 0.51 |
| | LPS (Ours) | 33.68 $\pm$ 0.96 | **61.58** $\pm$ 1.27 | **82.61** $\pm$ 0.39 | **91.72** $\pm$ 2.53 | **94.5** $\pm$ **0.80** |
| Trojan-WM (all-to-one) | Random | **6.75** $\pm$ 1.31 | 28.55 $\pm$ 3.17 | 61.06 $\pm$ 3.9 | 74.18 $\pm$ 0.42 | 80.74 $\pm$ 0.77 |
| | FUS | 6.35 $\pm$ 1.13 | 26.47 $\pm$ 6.56 | 51.05 $\pm$ 6.86 | 75.21 $\pm$ 3.15 | 85.34 $\pm$ 1.20 |
| | LPS (Ours) | 6.26 $\pm$ 0.81 | **48.26** $\pm$ 8.35 | **77.56** $\pm$ 2.93 | **86.16** $\pm$ 1.85 | **92.89** $\pm$ 1.72 |

Table 4: Results of various defenses against attacks on CIFAR-10. **Bold** means the best

| Attack | Defense | No Defense | | FT | | FP | | ABL | | NAD | | CLP | | I-BAU | |
|---|---|---|---|---|---|---|---|---|---|---|---|---|---|---|---|
| | | ASR | ACC | ASR | ACC | ASR | ACC | ASR | ACC | ASR | ACC | ASR | ACC | ASR | ACC |
| BadNets 0.216% (all-to-one) | Random | 69.73 | 93.97 | 35.82 | 93.87 | 3.88 | 93.43 | 18.86 | 62.42 | 1.18 | 88.23 | 9.57 | 92.91 | 2.18 | 84.64 |
| | FUS | 68.97 | 93.74 | 39.31 | 93.99 | 6.3 | 93.59 | 18.84 | 72.94 | 1.82 | 87.34 | 35.51 | 93.65 | 3.86 | 76.67 |
| | LPS (Ours) | **81.94** | 93.76 | **51.48** | 92.57 | **10.88** | 93.45 | **23.17** | 63.59 | **7.33** | 91.77 | **39.29** | 93.35 | **7.01** | 88.99 |
| Blended 0.216% (all-to-one) | Random | 53.22 | 94.01 | 33.26 | 93.85 | 24.39 | 93.47 | 30.07 | 71.75 | 23.58 | 91.59 | 32.53 | 93.33 | 9.77 | 76.32 |
| | FUS | 48.96 | 93.99 | 34.04 | 93.94 | 21.67 | 93.54 | 29.19 | 75.84 | 25.16 | 92.83 | **38.51** | 93.62 | 6.29 | 83.6 |
| | LPS (Ours) | **59.73** | 93.96 | **34.68** | 93.23 | **28.02** | 93.79 | **38.01** | 71.92 | **25.98** | 91.56 | 37.66 | 93.37 | 9.18 | 75.7 |
| SIG 0.216% (clean label) | Random | 12.61 | 93.86 | 12.58 | 93.59 | 10.84 | 93.45 | 13.99 | 73.69 | 2.08 | 90.88 | 15.48 | 93.63 | 2.99 | 87.26 |
| | FUS | 14.19 | 93.88 | 11.83 | 93.87 | 12.81 | 93.44 | 10.91 | 76.7 | 4.21 | 90.34 | 15.04 | 93.27 | 6.31 | 84.96 |
| | LPS (Ours) | **41.31** | 93.82 | **38.01** | 93.94 | **36.59** | 93.52 | **34.06** | 72.19 | **29.52** | 91.37 | **48.73** | 93.64 | **7.92** | 89.42 |
| Trojan-WM 0.216% (all-to-one) | Random | 89.43 | 93.73 | 86.4 | 93.6 | **46.59** | 93.15 | 51.71 | 71.5 | 43.21 | 91.2 | 2.74 | 92.75 | 7.29 | 84.57 |
| | FUS | 82.9 | 93.83 | 68.7 | 93.73 | 35.52 | 93.57 | 48.86 | 74.97 | 40.48 | 92.69 | 6.72 | 93.41 | **11.96** | 81.53 |
| | LPS (Ours) | **93.76** | 94.01 | **86.94** | 94.17 | 30.09 | 93.44 | **62.66** | 69.58 | **46.65** | 91.69 | **59.21** | 93.84 | 9.70 | 86.36 |

**Resistance to backdoor defense.** We further evaluate the resistance against defenses of different poisoning sample selection strategies. The defense results are shown in Tab. 4. It can be seen our method outperforms others in most cases (higher ASR is better), indicating that a reasonable poisoning sample selection strategy probably makes the attack better resistant to defenses.

**Resistance to noisy samples.** In the real world, the dataset may contain noisy samples, such as noisy labels and outliers. Hence, it is necessary to evaluate whether the poisoning sample selection strategies have the potential to be immune to such noisy samples. The results of resistance against noisy labels and outliers are shown in **Appendix** F.1. We find that LPS still outperforms other selection strategies, showing the robustness of LPS against noisy samples, demonstrating the practicality of LPS.

### 5.3 ABLATION STUDIES

**Effects of different constraints in LPS.** As demonstrated under Eq. (3), the equation $Hm = \tilde{\alpha} \cdot \mu$ captures three constraints, including satisfying the poisoning ratio, excluding the target class (dubbed ET), and selecting the same number of samples per class (dubbed PC), respectively. Here we compare LPS with its two variants of changing the last two constraints, in-

Table 5: Ablation studies of LPS's constraints.

| Attack | Pratio | LPS | LPS$\backslash_{ET}$ | LPS$\backslash_{ET,PC}$ | FUS |
|---|---|---|---|---|---|
| BadNets | 0.216% | 80.58 | 75.33 | 71.47 | 68.01 |
| Blended | 0.432% | 87.20 | 85.72 | 82.71 | 79.06 |
| SSBA | 0.432% | 23.29 | 21.18 | 20.36 | 14.86 |
| Trojan-WM | 0.216% | 93.27 | 89.91 | 87.80 | 77.63 |

cluding: **1)** *LPS without excluding target class* (LPS$\backslash_{ET}$), **2)** *LPS$\backslash_{ET}$ without selecting the same number of poisoned samples per class* (LPS$\backslash_{ET,PC}$). The results in Tab. 5 show that both constraints are important for the LPS strategy. Note that even removing two constraints, LPS$\backslash_{ET,PC}$ still outperforms FUS.

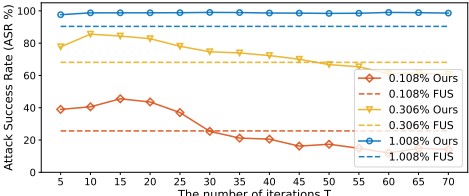 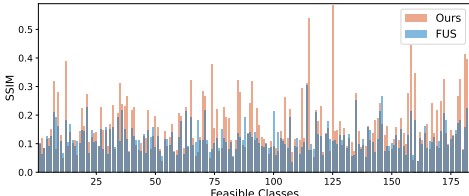

Figure 4: Attack results of LPS strategy on CIFAR-10 under different iterations $T$.

Figure 5: Average pairwise SSIM for each class computed over samples selected by our method and FUS on Tiny-ImageNet.

**Effect of the number of iterations $T$.** In Algorithm 1, our LPS method requires iteratively solving a min-max optimization problem. Here we explore the effect of different iterations $T$ on the attack results. As shown in Fig. 4, we evaluate LPS strategy in a wide range of iterations from 1 to 50. We can see that LPS strategy shows stable and high performance in the range $T \in [10, 20]$. Therefore, we choose $T = 15$ as the default setting of the main experiments.

**Effect of the training setting of the target model.** As we make no assumptions about the training process of the target model, we also investigate the impact of various training settings of the target model on the attack performance in the **Appendix** F. Our experiments demonstrate substantial improvements across different settings.

## 6 ANALYSIS

**Analysis of computational complexity.** LPS uses an iterative algorithm by alternately updating the surrogate model and the poisoning mask. Concerning the update of the surrogate model in each iteration, the complexity is $O(|\mathcal{D}|C(F + B))$, where $|\mathcal{D}|$ is the size of the training data, $F$ is the cost of the forward pass in a DNN model, $B$ is the cost of the backward pass (Rumelhart et al., 1986), and $C$ is the number of epochs. Regarding the update of the poisoning mask, it requires one forward pass for all training samples, making the complexity $O(|\mathcal{D}|F)$. Consequently, the overall complexity for LPS is $O(T|\mathcal{D}|((C + 1)F + KB))$, where $T$ is the number of iterations. A comparison of the computation times of various methods is provided in the **Appendix** F. As LPS doesn't necessitate retraining in each iteration, the inner loop is executed only once. Consequently, the calculation time for LPS is significantly lower than that of other sample selection methods.

**Analysis of selected samples.** We examine the distinctions between samples selected by different strategies. First, we provide visualizations of some benign samples selected by LPS and FUS from the Tiny-ImageNet dataset in **Appendix** D. The results reveal that LPS tends to prioritize samples with distinct patterns. We further calculate the average pairwise structural similarity index (SSIM) (Wang et al., 2004) in each class for the selected samples to evaluate inter-class similarity. As shown in Fig. 5, samples selected by LPS exhibit a higher degree of inter-class similarity. These findings collectively suggest that the samples selected by LPS possess more pronounced characteristics.

## 7 CONCLUSION AND FUTURE WORK

This work has explored an often overlooked step in data-poisoning based backdoor attacks, *i.e.*, selecting which benign samples to generate poisoned samples. We innovatively propose a learnable poisoning sample selection strategy that is formulated as a min-max optimization, where a binary poisoning mask and a surrogate model are learned together, to select hard poisoned samples that contribute more to the backdoor learning. Extensive results validate the effectiveness and efficiency of the proposed LPS strategy in boosting various data-poisoning based backdoor attacks.

**Limitations and future works**. In the case of an extremely low poisoning ratio, the improvement of LPS is limited, mainly due to the fact that the poisoning information of few poisoned samples with fixed triggers is insufficient to inject backdoor, no matter which samples are selected. It inspires that learning trigger and poisoning sample selection simultaneously may further enhance the backdoor attack, which will be explored in future. In addition, the proposed LPS strategy is specially designed for data-poisoning based backdoor attack. Developing the similar selection strategy for training controllable backdoor attack also deserves to be explored in future.

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

## A    OVERVIEW OF THE APPENDIX

The overall structure of the Appendix is listed as follows:

## B    PROOF OF SOLUTION OF INNER MAXIMIZATION OPTIMIZATION PROBLEMS

### B.1    THE DECOMPOSITION OF EQ. (5)

**Theorem 1** *Under the setting of our optimization algorithm, the solution of the optimization problem of Eq. (5) is equivalent to the solution of the optimization problem of Eq. (6)*

**Proof 1**

$$
\begin{aligned}
&\max_{\boldsymbol{m}\in\{0,1\}^{|\mathcal{D}|},\boldsymbol{Hm}=\tilde{\alpha}\cdot\boldsymbol{\mu}} \mathcal{L}(\boldsymbol{m};\boldsymbol{\theta}_s,\mathcal{D},\boldsymbol{\epsilon},g)\\
&=\max_{\boldsymbol{m}\in\{0,1\}^{|\mathcal{D}|},\boldsymbol{Hm}=\tilde{\alpha}\cdot\boldsymbol{\mu}} \frac{1}{|\mathcal{D}|}\sum_{i=1}^{|\mathcal{D}|}\Big[(1-m_i)\cdot\ell(f_{\boldsymbol{\theta}_t}(\boldsymbol{x}_i),y_i))+m_i\cdot\ell(f_{\boldsymbol{\theta}_t}(\tilde{\boldsymbol{x}}_i),y_t)\Big]\\
&=\max_{\boldsymbol{m}\in\{0,1\}^{|\mathcal{D}|},\boldsymbol{Hm}=\tilde{\alpha}\cdot\boldsymbol{\mu}} \frac{1}{|\mathcal{D}|}\sum_{i=1}^{|\mathcal{D}|}\Big[\ell(f_{\boldsymbol{\theta}_t}(\boldsymbol{x}_i),y_i))+m_i\cdot\Big(\ell(f_{\boldsymbol{\theta}_t}(\tilde{\boldsymbol{x}}_i),y_t)-\ell(f_{\boldsymbol{\theta}_t}(\boldsymbol{x}_i),y_i))\Big)\Big]\\
&=\max_{\boldsymbol{m}\in\{0,1\}^{|\mathcal{D}|},\boldsymbol{Hm}=\tilde{\alpha}\cdot\boldsymbol{\mu}} \frac{1}{|\mathcal{D}|}\sum_{i=1}^{|\mathcal{D}|}\Big[m_i\cdot\Big(\ell(f_{\boldsymbol{\theta}_t}(\tilde{\boldsymbol{x}}_i),y_t)-\ell(f_{\boldsymbol{\theta}_t}(\boldsymbol{x}_i),y_i))\Big)\Big]+\mathcal{C},
\end{aligned}
\tag{7}
$$

*where the last equation is because* $\mathcal{C}=\frac{1}{|\mathcal{D}|}\sum_{i=1}^{|\mathcal{D}|}\ell(f_{\boldsymbol{\theta}_t}(\boldsymbol{x}_i),y_i))$ *is constant with respect to* $\boldsymbol{m}$.

*For the condtion* $\boldsymbol{Hm}=\tilde{\alpha}\cdot\boldsymbol{\mu}$, *according to the definition of* $\boldsymbol{H}$, *it is easy to get that*

$$
\boldsymbol{Hm}=\tilde{\alpha}\cdot\boldsymbol{\mu}\leftrightarrow\sum_{i\in\{j|y_j=k\}}m_i=\tilde{\alpha}\mu_k\quad\forall k\in\mathcal{Y}=\{1,\ldots,K\}.
$$

*For each class k, we can decompose the objective function as follows:*

$$\max_{\boldsymbol{m}\in\{0,1\}^{|\mathcal{D}|},\boldsymbol{Hm}=\tilde{\alpha}\cdot\boldsymbol{\mu}} \frac{1}{|\mathcal{D}|}\sum_{i=1}^{|\mathcal{D}|}\Big[m_i\cdot\Big(\ell(f_{\boldsymbol{\theta}_t}(\tilde{\boldsymbol{x}}_i),y_t)-\ell(f_{\boldsymbol{\theta}_t}(\boldsymbol{x}_i),y_i))\Big)\Big]$$

$$=\max_{\boldsymbol{m}\in\{0,1\}^{|\mathcal{D}|},\boldsymbol{Hm}=\tilde{\alpha}\cdot\boldsymbol{\mu}}\sum_{k=1}^{K}\sum_{\{i|y_i=k\}}\frac{1}{|\mathcal{D}|}\Big[m_i\cdot\Big(\ell(f_{\boldsymbol{\theta}_t}(\tilde{\boldsymbol{x}}_i),y_t)-\ell(f_{\boldsymbol{\theta}_t}(\boldsymbol{x}_i),y_i))\Big)\Big]$$

$$=\max_{\substack{\boldsymbol{m}\ \in\ \{0,1\}^{|\mathcal{D}|}\\ \sum_{\{i|y_i=k\}}m_i=\tilde{\alpha}\mu_k\ \ \forall k\in\mathcal{Y}}}\sum_{k\in\mathcal{Y}}\sum_{\{i|y_i=k\}}\frac{1}{|\mathcal{D}|}\Big[m_i\cdot\Big(\ell(f_{\boldsymbol{\theta}_t}(\tilde{\boldsymbol{x}}_i),y_t)-\ell(f_{\boldsymbol{\theta}_t}(\boldsymbol{x}_i),y_i))\Big)\Big]$$

$$=\max_{\substack{\boldsymbol{m}^k\ \in\ \{0,1\}^{n_k}\\ \sum_i m_i^k=\tilde{\alpha}\mu_k\ \ \forall k\in\mathcal{Y}}}\sum_{k\in\mathcal{Y}}f_k(m^k),$$

*where we set $\boldsymbol{m}^k=\Big[m_i\mid y_i=k\Big]^T\in\{0,1\}^{n_k}$ which represents the vector of the mask for the class k and $f_k(\cdot)$ as:*

$$f_k(z)=\sum_j\frac{1}{|\mathcal{D}|}\Big[z_j\cdot\Big(\ell(f_{\boldsymbol{\theta}_t}(\tilde{\boldsymbol{x}}_{jk}),y_t)-\ell(f_{\boldsymbol{\theta}_t}(\boldsymbol{x}_{jk}),y_{jk}))\Big)\Big],$$

*where $(\boldsymbol{x}_{jk},y_{jk})$ is the j-th sample of the set $\mathcal{D}_k=\{(x,y)\mid y=k,(x,y)\in\mathcal{D}\}$ that is the sample with class k.*

*We can also simplify the constraints as follows:*

$$\sum_{\{i|y_i=k\}}m_i=\mathbf{1}_{n_k}^T\boldsymbol{m}^k,$$

*where $\mathbf{1}_{n_k}$ is the all-ones vector with $n_k$ dimension. Because each sample belongs to only one label, no variables are shared between $\boldsymbol{m}^k$.*

*Since the constraints can be decomposed to uncontacted equations, the above optimization objective function can also be decomposed into K terms. Consequently, we can get K independent sub-problems, as follows: $\forall k\in\{1,2,\dots,K\}$*

$$\max_{\boldsymbol{m}^k\in\{0,1\}^{n_k}}\sum_{j=1}^{n_k}\frac{1}{|\mathcal{D}|}\Big[m_j^k\cdot\Big(\ell(f_{\boldsymbol{\theta}_t}(\tilde{\boldsymbol{x}}_{jk}),y_t)-\ell(f_{\boldsymbol{\theta}_t}(\boldsymbol{x}_{jk}),y_{jk}))\Big)\Big],$$

$$\text{s.t. }\mathbf{1}_{n_k}^T\boldsymbol{m}^k=\tilde{\alpha}\cdot\mu_k$$

### B.2 THE SOLUTION OF EQ. (6)

Eq. (6) is a constrained 0-1 integer programming problem. We denote $\delta_i^k$ as $\ell(f_{\boldsymbol{\theta}_t}(\tilde{\boldsymbol{x}}_{ki}),y_t)-\ell(f_{\boldsymbol{\theta}_t}(\boldsymbol{x}_{ki}),y_{ki}))$ and simplify the optimization objective function to

$$\sum_{i=1}^{n_k}\frac{1}{|\mathcal{D}|}m_i^k\cdot\delta_i^k.$$

For the constraint $\mathbf{1}_{n_k}^T\boldsymbol{m}^k=\tilde{\alpha}\mu_k$, with regrad to $\boldsymbol{m}^k\in\{0,1\}^{n_k}$, the number of elements in vector $\boldsymbol{m}^k$ that equals to 1 is $\tilde{\alpha}\cdot\mu_k$. Therefore, the procedure for solving Eq. (6) is finding top-$(\tilde{\alpha}\cdot\mu_k)$ of $\delta_i^k$ and setting the corresponding $m_i^k$ to 1, which means for each class k we select the sample corresponding to top-$(\tilde{\alpha}\cdot\mu_k)$ of $\ell(f_{\boldsymbol{\theta}_t}(\tilde{\boldsymbol{x}}_{ki}),y_t)-\ell(f_{\boldsymbol{\theta}_t}(\boldsymbol{x}_{ki}),y_{ki}))$.

## C MODIFICATION OF LPS UNDER DIFFERENT POISING STRATEGIES

In Sec. 4, our LPS sample selection strategy is mainly divided into two modules: constructing a min-max optimization problem and optimizing this optimization problem. The constraint $\boldsymbol{Hm}=\tilde{\alpha}\cdot\boldsymbol{\mu}$ in 3 corresponds to the situation of poisoning samples under different poisoning strategies, that is, for different poisoning strategies, we first need to modify $\tilde{\alpha}$ and $\boldsymbol{\mu}$.

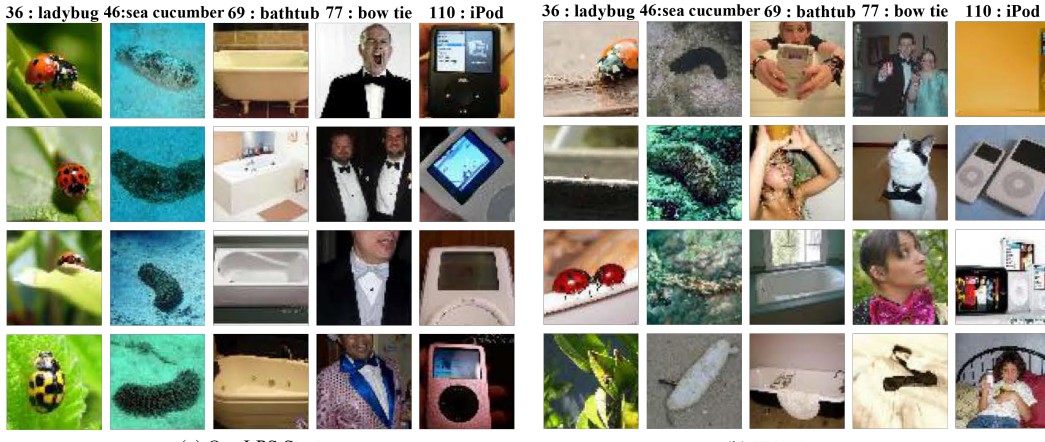

(a) Our LPS Strategy  (b) FUS Strategy

Figure 6: Visualization of samples selected by our LPS (a) and FUS (b).

- For all-to-all: Since all labels are target labels and each label is a source label, in optimization problem (3), $\tilde{\alpha}$ is changed to $\frac{\alpha*|\mathcal{D}|}{\sum_k n_k}$ and $\mu_k$ equals $n_k$ for any label k which means we poison all the labels evenly in the $\tilde{\alpha}$ ratio.
- For clean-label: Since the target label is still a source label and no other labels are poisoned, in optimization problem (3), $\tilde{\alpha}$ is changed to $\frac{\alpha*|\mathcal{D}|}{n_{y_t}}$, $\mu_k$ equals $n_k$ for target label $y_t$ and equals 0 for other label k which means we only poison data whose label is target label.

In the context of the outer minimization within the optimization process, no adjustments are necessary as the objective function remains unaltered. However, for the inner maximization during optimization, where modifications are made to both $\tilde{\alpha}$ and $\mu_k$ while leaving $H$ unaffected, the sole requirement is to perform a sorting algorithm on the non-zero indices of $\tilde{\alpha} \cdot \boldsymbol{\mu}$. Specifically, this involves sorting the losses associated with data labelled as k throughout the optimization process.

## D  VISUAL ANALYSIS

### D.1  VISUALIZATION OF SELECTED IMAGES

Fig. 6 shows the benign samples selected by different sample selection strategies, which is consistent with the phenomenon described in our manuscript.

### D.2  T-SNE OF POISONED DATASET

Appendix D.2 shows t-SNE embeddings about the poisoned dataset to analyze the latent space of backdoor models. We plot embeddings of 10% benign samples and all poisoned samples of the CIFAR-10 training set according to the features of layer4.1.conv2 of ResNet-34.

It shows that: **1)** The poisoned samples selected by LPS are more relatively scattered, while the samples selected by random selection and FUS are more likely to cluster together, especially at a low poisoning ratio. This phenomenon implies that it is more difficult for backdoor model to learn the backdoor mapping using the poisoned sample selected by LPS, i.e., the selected samples are more harder. **2)** From the defence perspective, the scattered poisoned samples exhibit more latent inseparability (Qi et al., 2022), thus may bypass some latent separation-based backdoor defenses.

### D.3  THE LOSS OF SELECTED IMAGES DURING TRAINING THE TARGET MODEL

In Fig. 8, we present the loss of poisoned images chosen through various strategies during the training of the target model, specifically ResNet34, on the contaminated dataset. It is observed that **1)** the

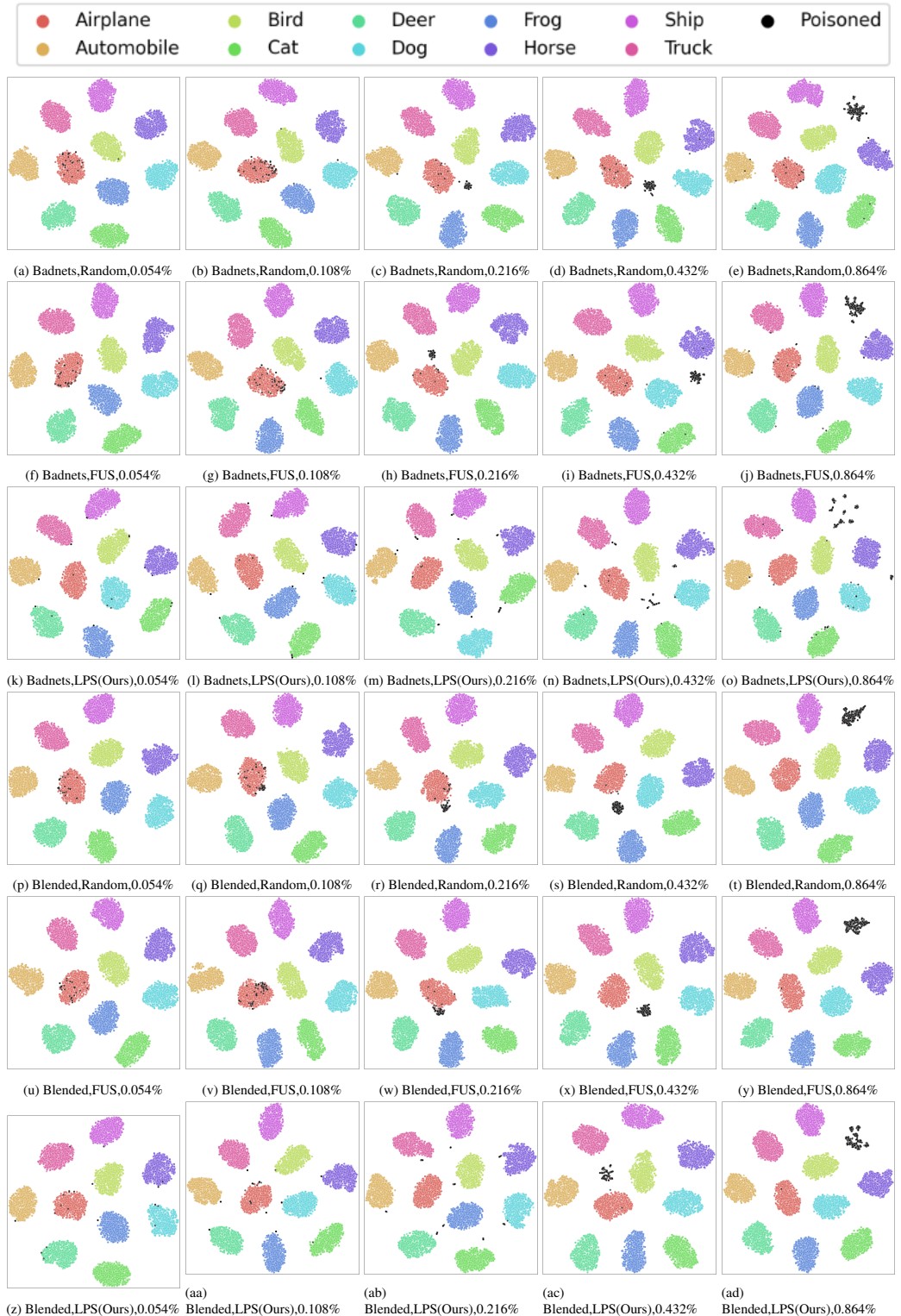

Figure 7: t-SNE visualization on CIFAR-10. Each point in the plots corresponds to a training sample. We randomly chose 10% benign samples and all poisoned samples for plotting, in which the poisoned samples are coloured in black.

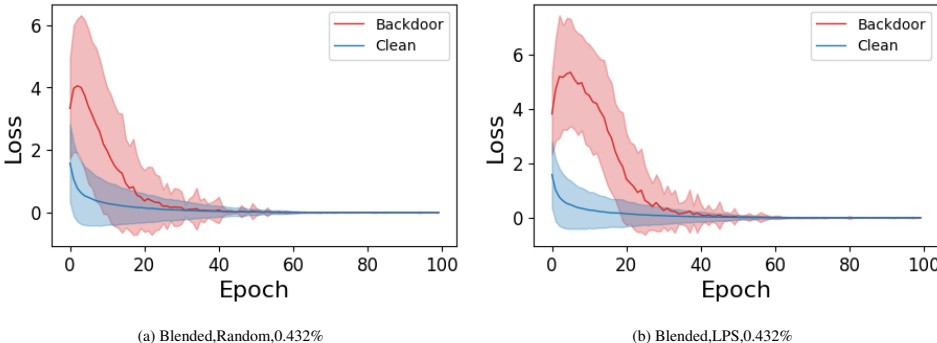

(a) Blended,Random,0.432%                    (b) Blended,LPS,0.432%

Figure 8: The loss of poisoned images selected by different strategies when the users train the target model on the poisoned dataset. The solid line denotes the mean loss value, while the dashed line spans an area two standard deviations wide.

loss of samples selected by our LPS strategy is comparatively higher than that of randomly chosen samples, and simultaneously, the convergence speed of LPS-selected poisoned samples is relatively slower. These phenomena suggest that the samples chosen by LPS pose a greater challenge for the target model to assimilate. **2)** In the later stages of the training process, the losses of samples selected by both methods converge almost to 0, indicating that the target model ultimately comprehensively learns these poisoned data.

In summary, the samples we selected exhibit a stronger poisoning effect, thereby enhancing the overall impact on the target model.

## E  IMPLEMENTATION DETAILS

### E.1  DETAILS OF BACKDOOR ATTACKS

In this section, we describe the general idea and detailed settings of 5 representative backdoor attack methods evaluated in the main manuscript. For all attacks, we use the codes implemented in BackdoorBench[2] Wu et al. (2022a).

- **BadNets** (Gu et al., 2019) is the pioneering study in the field of backdoor attacks, which replaces certain pixels in the original benign image with a visual patch to create a poisoned image. In our experiments, regardless of image sizes, we choose a $3 \times 3$ white square positioned 1 pixel away from the bottom right corner of the image as the trigger.

- **Blended** (Chen et al., 2017) firstly adopted the alpha blending strategy to fuse the trigger into the benign image. In our experiments, as the settings in Chen et al. (2017), we use a "Hello Kitty" cartoon image as a trigger, and the blend ratio is set as $0.15$.

- **SIG** (Barni et al., 2019) adds a sinusoidal signal designed by $v(i, j) = \Delta \sin(2\pi j f/m)$, for a certain frequency $f$, on the original image, where $m$ is the number of columns of the image and $l$ the number of rows. We set $\Delta = 20, f = 6$ for CIFAR-10 and CIFAR-100, $\Delta = 10, f = 6$ for Tiny-ImageNet.

- **Sample-specific backdoor attack (SSBA)** (Li et al., 2021b) adopted a double-loop auto-encoder to merge the string information into the benign image based on digital steganography, such that invisible and sample-specific triggers could be generated. The auto-encoder is trained for 140000 steps and the encoded bit is set as 1.

- **Trojaning Attack on Neural Networks (Trojan-WM)** (Liu et al., 2018b) inverse the neural network to optimise the trigger by maximizing its activation on selected neurons related. We use a watermark as the trigger shape and the trigger transparency is set as $85\%$.

- **All-to-one attack**: For all-to-one attacks, we set the target label of poisoning samples as 0.

- **All-to-all attack**: For all-to-all attacks, we set the target label of poisoning samples as the ground-truth label minus one, $e.g., 1 \Rightarrow 0, 2 \Rightarrow 1$.

[2]https://github.com/SCLBD/BackdoorBench/tree/main/attack

Table 6: Results of attack success rate with noise labels/outliers

| Attack | Settings | Noise Lable(ratio = 1%) | | | Noise Lable(ratio = 10%) | | | Outliers(ratio = 1%) | | | Outliers(ratio = 10%) | | |
|---|---|---|---|---|---|---|---|---|---|---|---|---|---|
| | Pratio | 0.216% | 0.432% | 0.864% | 0.216% | 0.432% | 0.864% | 0.216% | 0.432% | 0.864% | 0.216% | 0.432% | 0.864% |
| BadNets | Random | 55.96% | 80.53% | 88.68% | 3.62% | 81.4% | 87.4% | 70.72% | 83.75% | 87.88% | 64.49% | 84.64% | 89.7% |
| | LPS | 61.99% | 83.05% | 90.71% | 9.18% | 86.39% | 90.56% | 76.01% | 85.34% | 89.19% | 74.98% | 85.57% | 91.46% |
| Blended | Random | 45.76% | 68.33% | 84.8% | 35.88% | 63.91% | 82.32% | 47.19% | 73.42% | 88.28% | 46.45% | 69.24% | 86.03% |
| | LPS | 54.61% | 86.34% | 97.2% | 37.29% | 69.59% | 94.51% | 60.59% | 88.91% | 97.68% | 60.9% | 85.84% | 97.7% |

- **Clean label attack**: For clean label attacks, we set the target label of testing poisoning samples as 0, and do not change the trained poisoning samples' target label.

## E.2 Details of backdoor defenses

In this section, we describe the general idea and detailed settings of 6 representative backdoor defense methods evaluated in the main manuscript. For all defenses, we use the codes implemented in BackdoorBench[3] (Wu et al., 2022a).

- **Fine-tuning (FT)**: This defense fine-tune the backdoored model on a subset of clean samples to mitigate the backdoor. We use $5\%$ ratio of validation dataset to fine-tune with 100 epochs.

- **Fine-pruning (FP)** (Liu et al., 2018a): firstly prunes some inactivated neurons of clean samples with the maximum relative drop of clean accuracy is $10\%$. The ratio of validation data is $5\%$.

- **Anti-Backdoor Learning (ABL)** (Li et al., 2021a): isolates poisoned samples from benign samples according to their difference on loss dropping speed, then mitigates the backdoor effect by maximizing the loss of the isolated poisoned samples. The isolation ratio of training data is 0.01, the tuning epochs, fine-tuning epochs, unlearning epochs are set as 10, 60, 20, respectively.

- **Neural Attention Distillation (NAD)** (Li et al., 2020a): adopts fine-tuned model as a teacher, and fine-tunes the backdoored model again by encouraging the consistency of the attention representation between the new fine-tuned model and the teacher model. The teacher model fine-tunes with $5\%$ for 10 epochs, and $\beta_1, \beta_2, \beta_3$ of the loss are set as $500, 1000, 2000$, respectively.

- **Channel Lipschitzness Pruning (CLP)** (Zheng et al., 2022): proposes a Lipschitz constant of the mapping from the input images to the output of each channel to prune channels. The hyperparameter $u$ is defaulted to be 3 on CIFAR-10 and 5 on Tiny-ImageNet.

- **Implicit Backdoor Adversarial Unlearning (I-BAU)** (Zeng et al., 2021a): proposes a minimax formulation for removing backdoor from a given poisoned model based on a small set of clean data. The outer and inner iteration numbers are set as 5 and 1. The $\ell_2$-norm bound is 10.

## F Additional experiments

### F.1 Resistance to noise samples

**Setting** For noise labels, we replace their original labels with randomly assigned labels. For outliers, we blend such the chosen samples with other samples randomly chosen from the same classes. Blending ratio is set as 0.5.

It can be seen our methods achieve high ASR in Tab. 6. We find that among the samples selected by LPS, the proportion of noisy labels and outliers is small. LPS tend to select samples with high gap between backdoor loss $\ell(f_{\boldsymbol{\theta}_s}(\tilde{\boldsymbol{x}}_i), y_t)$ and clean loss $\ell(f_{\boldsymbol{\theta}_s}(\boldsymbol{x}_i), y_i)$ according to Eq.6. For noisy labels, the clean loss of these noisy samples is relatively large. For outliers, due to their different feature distribution, the backdoor mapping learned from these samples is hard to generalize to normal samples, which causes the loss gap of other samples to become larger. Therefore, LPS is minimally affected by these data points.

---

[3] https://github.com/SCLBD/BackdoorBench/tree/main/defense

### F.2 COMPARISON OF RUNNING TIME IN SECTION 6

In this section, we compare actual running time of different poisoning sample selection strategies under the same experimental environment. All experiments are performed on a machine with Intel Xeon 8170 CPU and NVIDIA GeForce RTX 3090 GPUs. Tab. 7 shows the average running time of Blended attack on three datasets with ResNet-18. It can be seen that our proposed LPS strategy saves about 82% of the running time compared to FUS. In the meantime, it only takes 27% more running time compared to random strategy to achieve a more effective attack, which is consistent with their computational complexities mentioned in Sec. 6 of the main manuscript.

Table 7: Average running time of Blended attack with ResNet-18.

|        | CIFAR-10  | CIFAR-100 | Tiny-ImageNet |
|--------|-----------|-----------|---------------|
| Random | 37m 29s   | 40m 41s   | 173m 23s      |
| FUS    | 264m 51s  | 295m 32s  | 1197m 12s     |
| Ours   | 47m 17s   | 51m 26s   | 206m 38s      |

### F.3 COMPARISON OF DIFFERENT MODEL ARCHITECTURES

To evaluate the generalization of our LPS strategy on various model architectures, we conduct external experiments using ResNet (He et al., 2016), VGG (Simonyan & Zisserman, 2015), MobileNet (Howard et al., 2017), and DenseNet (Huang et al., 2017).

#### F.3.1 SURROGATE: RESNET-18, TARGET: RESNET-18

When both the surrogate model and target model have the same architecture, chosen as ResNet-18 (He et al., 2016), the attack results are presented in Tab. 8. It is evident that the performance improvement achieved by our LPS strategy is more significant compared to when the target model is ResNet-34, as mentioned in the main manuscript.

#### F.3.2 SURROGATE: VGG11, TARGET: RESNET-18

Tab. 9 illustrates the attack results when the surrogate model is VGG11 (Simonyan & Zisserman, 2015) and the target model is ResNet-18 (He et al., 2016). These results indicate the applicability of our LPS strategy to various model architectures.

#### F.3.3 SURROGATE: MOBILENET-V2, TARGET: RESNET-18

The attack results presented in Tab. 10 correspond to the case where the surrogate model is MobileNet-v2 (Howard et al., 2017) and the target model is ResNet-18 (He et al., 2016). These results further validate the generalization ability of our LPS strategy.

#### F.3.4 SURROGATE: DENSENET-121, TARGET: RESNET-18

Tab. 11 presents the attack results obtained when the surrogate model is DenseNet-121 (Huang et al., 2017) and the target model is ResNet-18 (He et al., 2016). These results further confirm the generalization ability of our LPS strategy.

### F.4 COMPARISON OF DIFFERENT TRAINING HYPER-PARAMETERS

To further analyze how different hyper-parameters of the surrogate model and the target model will influence the attack results, we conduct external experiments using different optimizers(SGD (Bottou & Bousquet, 2007)/Adam (Kingma & Ba, 2014)) and parameters(batch-size and learning-rate).

#### F.4.1 COMPARISON OF DIFFERENT OPTIMIZERS

We conducted experiments under two distinct scenarios. In the first scenario, both the surrogate model and the target model employed the SGD optimizer, with a learning rate (lr) of 0.01 and a

Table 8: Attack success rate (%) on CIFAR-10, where the surrogate and target model are ResNet-18 and ResNet-18 respectively. **Bold** means the best.

| | **Dataset: CIFAR-10** | **Surrogate: ResNet-18 $\Longrightarrow$ Target: ResNet-18** | | | | |
|---|---|---|---|---|---|---|
| Attack | Pratio (#Img/Cls) | 0.00054 (#3) | 0.00108 (#6) | 0.00216 (#12) | 0.00432 (#24) | 0.00864 (#48) |
| BadNets (all-to-one) | Random | 0.9 | 2.88 | 61.84 | 82.39 | 90.86 |
| | FUS | 1.01 | 1.72 | 64.92 | 83.8 | **91.51** |
| | LPS (Ours) | **1.09** | **29.44** | **82.73** | **89.49** | 89.26 |
| BadNets (all-to-all) | Random | 0.75 | 0.78 | 1.08 | 40.53 | 73.15 |
| | FUS | 0.75 | 0.64 | 0.99 | 26 | 68.36 |
| | LPS (Ours) | **0.87** | **0.91** | **19.9** | **55.81** | **73.55** |
| Blended (all-to-one) | Random | 10.01 | 26.83 | 57.28 | 79.44 | 89.69 |
| | FUS | 13.88 | 24.19 | 55.71 | 83.32 | 93.4 |
| | LPS (Ours) | **10.34** | **43.8** | **71.47** | **89.53** | **98.29** |
| Blended (all-to-all) | Random | 2.25 | 3.27 | 9.75 | 40.03 | 70.79 |
| | FUS | 1.94 | 2.86 | 5.43 | 35.39 | 66.66 |
| | LPS (Ours) | **3.1** | **8.64** | **35.51** | **61.48** | **71.76** |
| SIG (clean label) | Random | 3.24 | 6.72 | 15.38 | 25.9 | 46.69 |
| | FUS | 2.34 | 9.66 | 15.14 | 21.76 | 40.23 |
| | LPS (Ours) | **12.07** | **31.66** | **45.99** | **60.34** | **66.83** |
| SSBA (all-to-one) | Random | 0.97 | **1.78** | 3.43 | 30.54 | 64.12 |
| | FUS | **1.21** | 1.48 | 2.97 | 23.71 | **69.48** |
| | LPS (Ours) | 0.99 | 1.52 | **5.37** | **30.61** | 68.5 |
| Trojan-WM (all-to-one) | Random | 3.42 | 21.6 | 87.4 | 96.27 | 98.84 |
| | FUS | **5.74** | 26.43 | 80.44 | 96.72 | 99.11 |
| | LPS (Ours) | 4.36 | **60.63** | **96.43** | **99.98** | **99.99** |

Table 9: Attack success rate (%) on CIFAR-10, where the surrogate and target model are VGG11 and ResNet-18 respectively. **Bold** means the best.

| | **Dataset: CIFAR-10 Surrogate: VGG11 $\Longrightarrow$ Target: ResNet-18** | | | | | |
|---|---|---|---|---|---|---|
| Attack | Pratio (#Img/Cls) | 0.00054 (#3) | 0.00108 (#6) | 0.00216 (#12) | 0.00432 (#24) | 0.00864 (#48) |
| BadNets (all-to-one) | Random | 0.9 | 2.88 | 61.84 | 82.39 | **90.86** |
| | FUS | 0.76 | 1.44 | 51.29 | 81.98 | 88.3 |
| | LPS (Ours) | **0.91** | **6.71** | **76.13** | **87.54** | 89.08 |
| BadNets (all-to-all) | Random | 0.75 | 0.78 | 1.08 | 40.53 | **73.65** |
| | FUS | **0.82** | 0.73 | 1.05 | 22.54 | 67 |
| | LPS (Ours) | 0.81 | **0.88** | **15.38** | **61.72** | 73.43 |
| Blended (all-to-one) | Random | 10.01 | 26.83 | 57.28 | 79.44 | 89.69 |
| | FUS | 11.97 | 25.77 | 53.82 | 80.83 | 92.37 |
| | LPS (Ours) | **15.69** | **43.16** | **70.22** | **85.63** | **95.17** |
| Blended (all-to-all) | Random | 2.25 | 3.27 | 9.75 | 40.03 | 70.79 |
| | FUS | 2.47 | 3.6 | 5.32 | 37.11 | 67.63 |
| | LPS (Ours) | **3.56** | **7.66** | **31.12** | **55.66** | **71.4** |
| SIG (clean label) | Random | 3.24 | 6.72 | 15.38 | 25.9 | 46.69 |
| | FUS | **12.01** | 19.87 | 72.68 | 91.71 | 97.01 |
| | LPS (Ours) | 11.39 | **59.34** | **85.31** | **93.06** | **98.26** |
| SSBA (all-to-one) | Random | 0.97 | **1.78** | 3.43 | 30.54 | 64.12 |
| | FUS | **1.21** | 1.48 | 3.21 | 28.22 | 67.1 |
| | LPS (Ours) | 1.2 | 1.52 | **5.61** | **35.99** | **68.18** |
| Trojan-WM (all-to-one) | Random | 3.42 | 21.6 | 87.4 | 96.27 | 98.84 |
| | FUS | 5.06 | 22.24 | 77.58 | 96.2 | 99.39 |
| | LPS (Ours) | **6.74** | **68.57** | **95.14** | **99.19** | **99.83** |

Table 10: Attack success rate (%) on CIFAR-10, where the surrogate and target model are MobileNet-v2 and ResNet-18 respectively. **Bold** means the best.

| | **Dataset: CIFAR-10** | **Surrogate: MobileNet-v2 $\Longrightarrow$ Target: ResNet-18** | | | | |
|---|---|---|---|---|---|---|
| Attack | Pratio (#Img/Cls) | 0.00054 (#3) | 0.00108 (#6) | 0.00216 (#12) | 0.00432 (#24) | 0.00864 (#48) |
| BadNets (all-to-one) | Random | 0.9 | 2.88 | 61.84 | 82.39 | **90.86** |
| | FUS | 0.9 | 1.3 | 41.18 | 81.13 | 89.54 |
| | LPS (Ours) | **1.16** | **15.94** | **76.46** | **88.02** | 90.31 |
| BadNets (all-to-all) | Random | 0.75 | 0.78 | 1.08 | 40.53 | 71.65 |
| | FUS | 0.84 | **1.56** | 1.19 | 9.54 | 61.89 |
| | LPS (Ours) | **0.85** | 0.96 | **8.14** | **59.08** | **71.72** |
| Blended (all-to-one) | Random | 10.01 | 26.83 | 57.28 | 79.44 | 89.69 |
| | FUS | 12.47 | 22.06 | 49.72 | 78.17 | 91.49 |
| | LPS (Ours) | **15.92** | **36.56** | **71.64** | **83.73** | **94.66** |
| Blended (all-to-all) | Random | 2.25 | 3.27 | 9.75 | 70.03 | **70.79** |
| | FUS | 3.17 | **8.07** | 5.27 | **76.91** | 68.39 |
| | LPS (Ours) | **4.95** | 7.2 | **21.93** | 75.73 | 67.59 |
| SIG (clean label) | Random | 3.24 | 6.72 | 15.38 | 25.9 | 46.69 |
| | FUS | 3.06 | 30.9 | 71.36 | 24.72 | 97.03 |
| | LPS (Ours) | **9.46** | **42.49** | **79.73** | **93.41** | **99.1** |
| SSBA (all-to-one) | Random | 0.97 | **1.78** | 3.43 | 30.54 | 64.12 |
| | FUS | **1.24** | 1.56 | 2.73 | 24.09 | 62.67 |
| | LPS (Ours) | 1.12 | 1.27 | **5.71** | **33.61** | **67.68** |
| Trojan-WM (all-to-one) | Random | 3.42 | 21.6 | 87.4 | 96.27 | 98.84 |
| | FUS | 6.1 | 27.04 | 62.36 | 95.3 | 99.38 |
| | LPS (Ours) | **6.3** | **50.28** | **96.63** | **98.82** | **99.61** |

Table 11: Attack success rate (%) on CIFAR-10, where the surrogate and target model are DenseNet-121and ResNet-18 respectively. **Bold** means the best.

| | **Dataset: Cifar10** | **Surrogate: DenseNet-121 $\Longrightarrow$ Target: ResNet-18** | | | | |
|---|---|---|---|---|---|---|
| Attack | Pratio (#Img/Cls) | 0.00054 (#3) | 0.00108 (#6) | 0.00216 (#12) | 0.00432 (#24) | 0.00864 (#48) |
| BadNets (all-to-one) | Random | 0.9 | 2.88 | 61.84 | 82.39 | **90.86** |
| | FUS | **1.09** | 1.84 | 48.51 | 79.94 | 89.93 |
| | LPS (Ours) | 0.9 | **10.72** | **84.81** | **86.36** | 88.84 |
| BadNets (all-to-all) | Random | 0.75 | 0.78 | 18.08 | 40.53 | 71.65 |
| | FUS | 1 | **1.93** | 20.52 | **68.71** | 70.3 |
| | LPS (Ours) | 0.89 | 1.91 | **21.29** | 66.05 | **72.27** |
| Blended (all-to-one) | Random | 10.01 | 26.83 | 57.28 | 79.44 | 89.69 |
| | FUS | 12.28 | 22.66 | 54.71 | 78.81 | 92.28 |
| | LPS (Ours) | **15.61** | **45.43** | **69.27** | **85.14** | **94.7** |
| Blended (all-to-all) | Random | 2.25 | 3.27 | 9.75 | 40.03 | 70.79 |
| | FUS | **10.21** | 27 | 31.46 | 57.21 | 69.8 |
| | LPS (Ours) | 4.07 | **8.86** | **32.43** | **59.5** | **71.49** |
| SIG (clean label) | Random | 3.24 | 6.72 | 15.38 | 25.9 | 46.69 |
| | FUS | 3.09 | 25.41 | 80.34 | 33.02 | 97.82 |
| | LPS (Ours) | **13.44** | **49.56** | **79.93** | **93.23** | **98.89** |
| SSBA (all-to-one) | Random | 0.97 | **1.78** | 3.43 | 30.54 | 64.12 |
| | FUS | **1.19** | 1.63 | 2.86 | 23.52 | 63.23 |
| | LPS (Ours) | 1.04 | 1.33 | **4.92** | **35.09** | **65.38** |
| Trojan-WM (all-to-one) | Random | 3.42 | 21.6 | 87.4 | 96.27 | 98.84 |
| | FUS | 4.76 | 21.06 | 69.13 | 97.14 | 99.1 |
| | LPS (Ours) | **7.27** | **58.42** | **93.41** | **99.38** | **99.96** |

weight decay of 5e-4. In the second scenario, the surrogate model utilized the Adam optimizer with default parameters (betas set to [0.9, 0.999] and lr set to 0.01), while the target model continued to use the SGD optimizer with the same lr and weight decay settings. Tab. 12 shows discrepancies between these two scenarios, which exhibit the difference between the optimizers used in the surrogate model and the target model resulting in slightly reduced ASR. This indicates that the choice of optimizers has a minor impact on our LPS strategy.

Table 12: Attack success rate (%) on CIFAR-10, where the surrogate and target model use different optimizers. **Bold** means the best.

| | Dataset: CIFAR-10 | | |
|---|---|---|---|
| Attack | Pratio (#Img/Cls) | 0.00216 (#12) | 0.00432 (#24) |
| BadNets | Random | 62.57 | 81.71 |
| | LPS(SGD$\Rightarrow$ SGD) | **76.41** | **85.77** |
| | LPS(Adam$\Rightarrow$ SGD) | 74.81 | 83.7 |
| Blended | Random | 50.65 | 75.67 |
| | LPS(SGD$\Rightarrow$ SGD) | **64.6** | **87.16** |
| | LPS(Adam$\Rightarrow$ SGD) | 61.24 | 85.93 |

### F.4.2 COMPARISON OF DIFFERENT PARAMETERS

We evaluate the generalization ability of our LPS method when the surrogate model and the target model use different training parameters. Although we find that the training parameters can have a significant influence on backdoor attacks, Tabs. 13 and 14 demonstrates a noteworthy enhancement achieved through our LPS approach.

Table 13: Attack success rate (%) on CIFAR-10, where the surrogate and target model use different batch sizes. **Bold** means the best.

| | Dataset: CIFAR-10 Pratio:0.43% batch size = 128 | | |
|---|---|---|---|
| Attack | Victim model | batch size = 64 | batch size = 128 | batch size = 256 |
| BadNets | Random | 88.35 | 81.71 | 76.04 |
| | LPS | **92.37** | **85.77** | **81.6** |
| blended | Random | 77.7 | 75.6 | 70.36 |
| | LPS | **92.41** | **87.16** | **80.97** |

Table 14: Attack success rate (%) on CIFAR-10, where the surrogate and target model use different learning rates(lr). **Bold** means the best.

| | Dataset: CIFAR-10 Pratio:0.43% lr=0.01 | | |
|---|---|---|---|
| Attack | Victim model | lr=0.006 | lr=0.01 | lr=0.014 |
| BadNets | Random | 80.57 | 81.71 | 85.59 |
| | LPS | **82.26** | **85.77** | **86.84** |
| blended | Random | 73.74 | 75.6 | 76.43 |
| | LPS | **84.06** | **87.16** | **90.06** |

