# OpenReview forum: "Boosting Backdoor Attack with A Learnable Poisoning Sample Selection Strategy"
_ICLR.cc/2024/Conference — Submitted to ICLR 2024_

### Official Review · Reviewer_i4rE · 2023-10-25

**Soundness:** 3 good
**Presentation:** 3 good
**Contribution:** 3 good
**Rating:** 5
**Confidence:** 3

**Summary:**

This paper points out a common weakness of most existing backdoor attacks, which is randomly choosing samples from the benign dataset to poison without considering how important different samples are. Instead of random selection, this paper proposes a novel strategy to choose poisoned data more efficiently by formulating it as a min-max optimization problem, called learnable poisoning sample selection strategy (LPS). Extensive experiments show that this strategy can improve data poisoning attacks with low poisoning rates.

**Strengths:**

1. The idea of improving backdoor attacks' efficiency by having better sample selection to inject backdoors is intriguing.
2. The proposed method is quite interesting. By effectively choosing highly influential poisoned samples, it allows different types of backdoor attack to reach high ASRs with very few number of poisoned samples. It can also overcome the limitations of previous related work (FUS).
3. The paper includes vastly extensive experiments to show the effectiveness of the proposed method.

**Weaknesses:**

1. I find the results of this method with SSBA are quite underwhelming: in most experiments with SSBA, LPS could not reach high ASRs and seem to not have much improvement compared to the baselines. Therefore, I am not sure about LPS's flexibility, i.e., whether it can work with any type of backdoor trigger.
2. Although there are experiments with different model architectures for the surrogate model, only ResNet is used for the target model. In the paper's settings, the adversary have full control of the process of generating poisoned samples but no control in the victim model's training procedure, so I think it would make more sense if there were experiments with different victim model's architectures.
3. The experimental results of LPS's resistance against backdoor defenses are quite unsatisfying: in the cases of FP, ABL, NAD, and I-BAU, the ASRs are greatly degraded and/or have merely limited improvement compared to other baselines.  Also, there are experiments with 6 backdoor defenses, but none of them are data filtering defense, such as [1], [2], [3], [4]. Since the adversary in this paper acts as data provider, I think there should be also evaluations with data filtering defenses.


[1] Chen, Bryant, et al. "Detecting Backdoor Attacks on Deep Neural Networks by Activation Clustering." (AAAI 2019)
[2] Tran, Brandon, Jerry Li, and Aleksander Madry. "Spectral signatures in backdoor attacks." (NeurIPS 2018)
[3] Hayase, Jonathan, et al. "Spectre: Defending against backdoor attacks using robust statistics." (ICML 2021)
[4] Zeng, Yi, et al. "Rethinking the backdoor attacks' triggers: A frequency perspective." (ICCV 2021)

**Questions:**

1. Could the authors provide any insights/explanations for LPS's low performances with SSBA? Could LPS work with any attack method, or the adversary should carefully choose a suitable type of trigger?
2. Regarding my concerns about victim model's architecture and data filtering defenses, I would recommend adding aforementioned experiments.
3. All datasets used in this work have relatively low resolution. Could this method also work with high resolution datasets, such as CelebA, PubFig, ImageNet?

---

> ### Author Response · Authors · 2023-11-17
> **Response to Reviewer i4rE (Part 1)**
>
> Dear Reviewer i4rE,
>
> We sincerely appreciate your precious time and constructive comments, and we are greatly encouraged by your high recognition of the our **intriguing idea**, **effective approach** and **extensive experiments**.
>
> In the following, we would like to answer your concerns separately.
>
> ---
>
> **Q1:** I find the results of this method with SSBA are quite underwhelming: in most experiments with SSBA, LPS could not reach high ASRs and seem to not have much improvement compared to the baselines. Therefore, I am not sure about LPS's flexibility, i.e., whether it can work with any type of backdoor trigger. Could the authors provide any insights/explanations for LPS's low performances with SSBA? Could LPS work with any attack method, or the adversary should carefully choose a suitable type of trigger?
>
> **R1:** Thank you for this insightful comment. For the performance of LPS on SSBA, although the improvement on CIFAR-10 is relatively low, **LPS significantly improves attack on Tiny-ImageNet dataset**. We would like to explain from the following aspects:
>
> - **The improvement of attack by LPS stargety depends on the strength of the underlying attack.**  In instances where the poisoning ratio is exceedingly low and the attack is relatively weak, injecting the backdoor into the model via outer minimization optimization becomes challenging. Consequently, the inner maximization optimization is hard to learn a reasonable mask to select samples.
> - **SSBA attack is weak on low-resolution dataset with low poisoning ratio.** SSBA generates invisible additive noises as triggers by encoding an attacker-specified string into benign images through an encoder-decoder network. Hence, it's hard to hide such information at very low-resolution images (32*32) in CIFAR-10 dataset, resulting in an insignificant attack effect at a low poisoning ratio.
>
> ---
>
> **Q2:** Although there are experiments with different model architectures for the surrogate model, only ResNet is used for the target model. In the paper's settings, the adversary have full control of the process of generating poisoned samples but no control in the victim model's training procedure, so I think it would make more sense if there were experiments with different victim model's architectures. Regarding my concerns about victim model's architecture and data filtering defenses, I would recommend adding aforementioned experiments.
>
> **R2:** Thanks for this constructive suggestion. **We additionally evalute our method on the popular transformer based-model (vit_b_16)** when the surrogate model is convolution based-model (ResNet18). The results are shown in Table 1 as follows. From the results, we can find that our LPS strategy is still effective when applied to transformer based-model.
>
> Table 1: Attack results on CIFAR-10, where surrogate model is ResNet18 and target model is Vit_b_16.
> |Pratio=0.216%| BadNets (ASR %) | Blended (ASR %) |
> | -------------- | --------------- | --------------- |
> | Random         |39.9|72.47|
> | **LPS (Ours)** |43.42|76.2|
>
> ---

---

> ### Author Response · Authors · 2023-11-17
> **Response to Reviewer i4rE (Part 2)**
>
> ---
>
> **Q3:** All datasets used in this work have relatively low resolution. Could this method also work with high resolution datasets, such as CelebA, PubFig, ImageNet?
>
> **R3:** Thanks for this constructive suggestion.
>
> **We evaluate our proposed LPS strategy on full ImageNet-1k with 1000 classes and higher resolution. (224*224)** Specifically, we choose two samples from each class for poisoning and the poisoning ratio is 0.156%. The results are shown in Table 2 as follows. The results demonstrate that our LPS strategy is effective for different resolutions.
>
> Table 2: Attack results on ImageNet-1k, where surrogate model is ResNet18 and target model is ResNet34
> |Pratio=0.156%| BadNets (ASR %) | Blended (ASR %) |
> | -------------- | --------------- | --------------- |
> | Random         |81.3|90.21|
> | **LPS (Ours)** |84.02|92.78|
>
> ---
>
> **Q4:** The experimental results of LPS's resistance against backdoor defenses are quite unsatisfying: in the cases of FP, ABL, NAD, and I-BAU, the ASRs are greatly degraded and/or have merely limited improvement compared to other baselines. Also, there are experiments with 6 backdoor defenses, but none of them are data filtering defense. Since the adversary in this paper acts as data provider, I think there should be also evaluations with data filtering defenses.
>
> **R4:** Thanks for this construstive comment. **We additionally evaluate on four poisoned sample detection methods on CIFAR-10, including  FREAK[1], Spectral[2], AC[3], Strip[4].** The results are shown in Table 3 as follows, where TPR denotes the proportion of selected poisoned samples to all poisoned samples. The results show that LPS ahieves similar performance compared to random selection, indicating that LPS does not increase the risk of being deteced.
>
> Table 3: Results of various backdoor detection method against attacks on CIFAR-10, where poisoning ratio is set as 0.108%.
> |                  | FREAK (TPR%) | Spectral (TPR%) | AC (TPR%) | Strip (TPR%) |
> | ---------------- | ----------- | -------------- | -------- | ----------- |
> | Random (BadNets) | 11.11 | 0    | 0| 0|
> | LPS (BadNets)    | 3.70  | 0    | 0| 0|
> | Random (Blended) | 7.41  | 7.41 | 0| 0|
> | LPS (Blended)    | 9.26  | 7.41 | 0| 0|
>
> Thanks again for your time and attention. We hope the response can address your concerns.
>
> Best regards,
>
> Authors
>
> ---
>
> **Reference:**
> [1] Don’t FREAK Out: A Frequency-Inspired Approach to Detecting Backdoor Poisoned Samples in DNN. In CVPR 2023
>
> [2] Spectre: Defending against backdoor attacks using robust statistics. In ICML 2021
>
> [3] Detecting Backdoor Attacks on Deep Neural Networks by Activation Clustering. In AAAI 2019
>
> [4] STRIP: A Defence Against Trojan Attacks on Deep Neural Networks. In ACSAC 2019

---

> ### Author Response · Authors · 2023-11-21
> **Anticipating Your Feedback**
>
> Dear Reviewer i4rE,
>
> We would like to express our sincere gratitude for your valuable insights and suggestions on our work.
>
> We have tried our best to address the concerns and queries you raised during the rebuttal process. However, we would greatly appreciate knowing whether our response have effectively resolved your doubts. Your feedback will be instrumental in improving the quality of our work.
>
> We understand the demands of your busy schedule, and we genuinely value your contribution to the improvement of our manuscript. As the end of the discussion period is approaching, we eagerly await your reply before the end.
>
> Thank you once again for your time and attention.
>
> Best regards,
>
> Authors

---

> ### Author Response · Authors · 2023-11-22
> **Looking forward to your reply!**
>
> Dear i4rE,
>
> We would like to express our gratitude again for your valuable suggestions. As the deadline of ICLR rebuttal period is approaching, we look forward to hearing your feedback on our response. We would be grateful if you could re-evaluate our paper based on our response and new supplementary results.
>
> Best regards,
>
> Authors

---

### Official Review · Reviewer_mh2m · 2023-10-31

**Soundness:** 3 good
**Presentation:** 3 good
**Contribution:** 2 fair
**Rating:** 5
**Confidence:** 4

**Summary:**

In this paper, the authors study the data-poisoning-based backdoor attack. They propose to select the hard backdoor samples with the larger training loss on the surrogate model and formulate it as a min-max optimization problem. Extensive experiments show that it's better than the random selection and existing baseline.

**Strengths:**

1. It's an interesting idea to exploit hard examples to inject backdoors.
2. Experiments show it can improve existing backdoor attacks.
3. Code is provided.

**Weaknesses:**

1. It seems not robust against existing defenses according to Table 4. Although it improves the robustness of existing attacks, it's still defeated by existing defenses.
2. Some other attack and defense methods may be evaluated as well, such as [FTrojan](https://dl.acm.org/doi/abs/10.1007/978-3-031-19778-9_23), [ABS](https://dl.acm.org/doi/10.1145/3319535.3363216), [Unicorn](https://arxiv.org/abs/2304.02786).
3. Because it selects training samples with larger losses, it may be easily detected by scanning the dataset.
4. It would be better to show how to formulate the strategy for all-to-all and clean labels in the appendix.
5. It's good to show the performance with different poisoning rates. However, in the tables, most of the ASRs are lower than 90%. That means they are all unsuccessful attacks and not that useful. One may want to show a different ratio range.

**Questions:**

1. Can it be detected by scanning the dataset and selecting the outliers with respect to the training loss?
2. How does the robustness look like if the ASRs are above 90%? Currently, most of the ASRs in Table 4 are very low.
3. Are the losses for the selected samples aligned well on the surrogate model and the target model? That is if the target model also considers them as hard examples. Also, how does the loss change for those hard backdoor samples? Because existing research shows the backdoor features are usually easier to learn and thus have smaller losses.

---

> ### Author Response · Authors · 2023-11-19
> **Response to Reviewer mh2m (Part 1)**
>
> Dear Reviewer mh2m,
>
> We sincerely appreciate your precious time and constructive comments, and we are greatly encouraged by your high recognition of the our **interesting idea**, **comprehensive experiments**, and **published code**.
>
> In the following, we would like to answer your concerns separately.
>
> ---
>
> **Q1:** It seems not robust against existing defenses according to Table 4. Although it improves the robustness of existing attacks, it's still defeated by existing defenses.
>
> **R1:** Thanks for this constructive comment. **The performance of backdoor attacks are reflected in two aspects: (1)The effectiveness of attacks without defense.  (2)The resistance to defenses.**
> - **On the one hand, this paper focuses on boosting the attacks' effectiveness by proposing well-designed selection stragety.** The comprehensive experiments in the manuscript demonstrate that LPS indeed improves attacks' effectiveness.
> - **On the other hand, the resistance to defenses depends more on triggers**. However,  we do not make any changes to triggers of existing attacks. Hence, the cababilities of exising attacks to against defenses have not been weakened, which can be demonstrated as follows:
>     - **For existing backdoor defenses, we achieve similar performance reduction compared to random selection** (see Table 4 in manuscript). It is worth noting that since we have improved the effectiveness of the attacks unser no defense, the attacks are more robust to defenses.
>     - In **20 out of 24 experiments** of Table 4 in the manuscript, **our LPS strategy notably enhances the robustness of attacks against most defenses.** This underscores the effectiveness of a well-designed selection strategy in bolstering the attack's robustness against defenses.
> - In order to improve defense capabilities, we need to focus on designing more stealthy triggers combined with well-designed selection strategies, which remains to be further explored.
>
> ---
>
> **Q2:** Some other attack and defense methods may be evaluated as well, such as FTrojan, ABS, Unicorn.
>
> **R2:** Thanks for your suggestion. **We additionally evaluate on FTrojan[1], ABS[2] and Unicorn[3] that you suggested.**
> - We add comparison of FTrojan[1] attack on CIFAR-10. The results on FTrojan attack are shown in **Table 1**. The results show that our LPS is effective for FTrojan attack.
> - We add comparison of ABS[2] and Unicorn[3] detection methods on CIFAR-10 and the results are shown in **Table 2.** Due to the backdoor attacks with LPS stragegy are stronger than these with random selection, the backdoored models trained with LPS stragegy are easier to be detected by ABS and Unicorn.
>
> Table 1: Attack results of FTrojan, where surrogate model is ResNet18 and target model is ResNet34.
> |Method| Pratio | FTrojan（ASR%） |
> | ------- | ------ | ----------- |
> | Random | 0.216% | 93.87|
> | **LPS (Ours)** | 0.216% | 95.01|
>
> Table 2: Detection results of ABS and Unicorn, where surrogate model is ResNet18 and target model is ResNet34.
> |Method| ABS (Detection ACC %) | Unicorn (Detection ACC %) |
> | ---------- | --------------------- | ------------------------- |
> | Random     | 53.33| 86.67|
> | LPS (Ours) | 61.33| 90.67|
>
> ---

---

> ### Author Response · Authors · 2023-11-19
> **Response to Reviewer mh2m (Part 2)**
>
> **Q3:** Can it be detected by scanning the dataset and selecting the outliers with respect to the training loss?
>
> **R3:** Thanks for this comment. We would like to clarify this concern from the following aspects:
>
> - **Selecting poisoned samples by the attacker and training the target model by the user are two different stages (see Fig 1 in manuscript).** The selection process is inaccessible for the user and defenser. While the poisoned samples are selected based on loss gap, the crafted poisoned dataset are used for training a backdoored model by using standard cross-entropy loss. Therefore, the loss of backdoor samples will decrease as training progresses.
> - **We further empirically show the training loss of poisoned samples selected by random selection and LPS respectively in Table 2 as follows** (The detailed figure can be found in the revised manuscript). We find that the loss of samples selected by LPS is relatively larger than random selection in the early stage, which aligns with our purpose of identifying hard examples to enhance attack performance. However, the loss of items selected by both methods nearly converges to 0, resulting in the difficulty of detecting the poisoned sample by loss gap.
> - **Moreover, we add more comparisons to demonstrate that the our LPS does not increase the risk of being detected:**
>     - **For ABL defense[4] based on the observation that poisoned samples are more easily to be learned, we achieve similar performance reduction compared to random selection** (see Table 4 in manuscript). It is worth noting that since we have improved the effectiveness of the attacks unser no defense, the attacks are more robust to defenses.
>     - **Additionally, we add comparison on four poisoned sample detection mehtods**, including FREAK[5], Spectral[6], AC[7], Strip[8], and the results are shown in **Table 2** as follows. We find that LPS and random selection shows similar performance on TPR, means **our LPS does not increase the risk of being deteced.**
>     - **Furthermore, we  evaluate on BHN[9], which is a noise detection method**  based on the assumption that the cross-entropy loss of noise sample will be larger than that of normal sample. The results are shown in **Table 3** as follows. We find that the TPR of detecing BadNets is relatively low, indicating that **noise detector is hard to detect poisoned samples selected by our LPS strategy.**
>
> Table 2: The cross-entropy loss of poisoned samples selected by different strategies on CIFAR-10, where poisoning ratio is set as 0.432% and the target model is ResNet34.
> | Stage          | 0      | 5      | 10     | 15     | 20     | ...  | 75     | 80     | 85     | 90     | 95     |
> | -------------- | ------ | ------ | ------ | ------ | ------ | ---- | ------ | ------ | ------ | ------ | ------ |
> | Random         | 3.3336 | 3.3865 | 1.9746 | 1.9746 | 0.3659 | ...  | 0.0011 | 0.0002 | 0.0008 | 0.0008 | 0.0004 |
> | **LPS (Ours)** | 3.8211 | 5.3534 | 4.4994 | 3.1894 | 1.4226 | ...  | 0.0013 | 0.0053 | 0.0009 | 0.0012 | 0.0006 |
>
> Table 3: Detection results of BHN on CIFAR-10, where poisoning ratio is set as 0.216%.
> | Method         | BadNets (TPR %) | BadNets (FPR%) |
> | -------------- | --------------- | -------------- |
> | Random         | 11.12           | 2.65           |
> | **LPS (Ours)** | 11.93           | 2.72           |
>
> Table 4: Results of various backdoor detection method against attacks on CIFAR-10.
>
> |                  | FREAK (TPR%) | Spectral (TPR%) | AC (TPR%) | Strip (TPR%) |
> | ---------------- | ----------- | -------------- | -------- | ----------- |
> | Random (BadNets) | 11.99       | 22.73          | 57.36    | 27.92       |
> | LPS (BadNets)    | 10.28       | 10.32          | 37.08    | 31.34       |
> | Random (Blended) | 13.06       | 44.68          | 39.91    | 1.94        |
> | LPS (Blended)    | 10.37       | 28.01          | 19.86    | 2.08        |
>
> ---
>
> **Q4:** It would be better to show how to formulate the strategy for all-to-all and clean labels in the appendix.
>
> **R4:**  Thank you for your suggestion. We briefly describe it  here. More detailed descriptions are updated in the revised version highlighted with blue.
>
> For the all-to-all and clean label poisoning strategy, we only need to modify our optimization problem settings.
>
> - **For all-to-all attack:** Since all labels are target labels and each label is source label, in optimization problem (3), $\tilde{\alpha}$ is changed to $\frac{\alpha*|\mathcal{D}|}{\sum_{k}n_{k}}$, $\mu_{k}$ equals $n_{k}$ for any label k, which means we poison all the labels evenly with $\tilde{\alpha}$ ratio.
> - **For clean label attack:** Since the target label is still a source label and no other labels are poisoned, in optimization problem (3), $\tilde{\alpha}$ is changed to $\frac{\alpha*|\mathcal{D}|}{n_{y_{t}}}$, $\mu_{k}$ equals $n_{k}$ for target label $y_{t}$ and equals $0$ for other label k, which means we only poison samples whose label are target label.
>
> ---

---

> ### Author Response · Authors · 2023-11-19
> **Response to Reviewer mh2m (Part 3)**
>
> **Q5:** It's good to show the performance with different poisoning rates. However, in the tables, most of the ASRs are lower than 90%. At the same time, although the sample selection method can improve the robustness, most of the ASRs in Table 4 are lower than 90% too. That means they are all unsuccessful attacks and not that useful.
>
> **R5:** Thank you for your comments. We would like to clarify from the following aspects:
>
> - **This paper focuses more on improving the effectiveness of attacks at low poisoning ratios by applying selection strategy.** Most attacks at  high concentrations is already strong enough and the improvement are not obvious， while attacks at low ratios are less powerful and more likely to bypass  human inspection.
> - **Furthermore, we conducted evaluations at relatively higher poisoning ratios, specifically 5.4%, and 9%.** The corresponding results are presented in **Table 5**, proving that LPS is effective across different poisoning ratios.
> - **Our LPS achieves significant enhancements compared to baselines.** We examine the attack results on CIFAR-10 (see Table 1 in the manuscript), and the distribution of relative improvement rates compared to random selection are detailed in **Table 6** as follows. We find that our LPS achieves more than 10% improvement compared to random selection in 57.14% of cases. Notably, 20% of results accessed by LPS even achieve a relative improvement exceeding 100%. In contrast, for FUS, only 20.04% of the results achieved a relative improvement surpassing 10%.
> - **The improvement of selection strategy is affected by the attack method.** It is essential to underscore that LPS can be effortless integrated into any data-poisoning based backdoor attack. This adaptability makes our method more robust if stronger attacks are proposed in the future.
>
> Table 5: Attack results on CIFAR-10 with 5.4% and 9% poisoning ratios.
> || Pratio=5.4% | Pratio=9% |
> | - | - | - |
> | Random  (BadNets) | 93.57%      | 96.53%    |
> | LPS (BadNets)     | 94.21%      | 97.36%    |
> | Random  (Blended) | 94.7%       | 95.66%    |
> | LPS (Blended)     | 96.48%      | 97.54%    |
>
> Table 6: Distribution of relative improvement rate of attack results compared with random strategy
> | Relative improvement rate | Proportion (LPS) | Proportion (FUS) |
> | :-: | :-: | :-: |
> | < 1%| 8.57%            | 51.43%           |
> | 1% ~ 5%| 17.14%           | 22.86%           |
> | 5% ~ 10%| 17.14%           | 5.71%            |
> | 10% ~ 50%| 25.71%           | 17.14%           |
> | 50% ~ 100%| 11.43%           | 2.9%             |
> | > 100%| 20%              | 0%               |
>
> ---
>
> **Q6:** Are the losses for the selected samples aligned well on the surrogate model and the target model? That is if the target model also considers them as hard examples. Also, how does the loss change for those hard backdoor samples? Because existing research shows the backdoor features are usually easier to learn and thus have smaller losses.
>
> **R6:** Thank you for your comments. We would like to analyze it from the following perspectives.
>
> - **The poisoned samples selected by LPS strategy are also considered as hard examples for target model.** We have shown the cross-entropy loss change of selected poisoned samples for target model. (see **Table 2 in R3**) and the detailed **figure 8** is updated in the revised manuscript.
> - From the results, we find that **the loss of samples selected by LPS is relatively larger than random selection in the early stage**. Moreover, compared to random selection, **the convergence speed of such poisoned samples selected by LPS is also relatively slower**, indicaing that such samples are hard to learn. These phenomena aligns with our purpose of identifying hard examples to enhance attack performance.
> - **In the later stages of training process, the losses of these items selected by both methods  converge almost to 0**, demonstrating that the model eventually learned backdoor from these selected samples.
>
> ---
>
> Thanks again for your time and attention. We hope the response can address your concerns.
>
> Best regards,
>
> Authors
>
> ---
>
> **Reference:**
>
> [1] An Invisible Black-Box Backdoor Attack Through Frequency Domain. In ECCV 2022
>
> [2] ABS: Scanning Neural Networks for Back-doors by Artificial Brain Stimulation. In CCS 2019
>
> [3] UNICORN: A Unified Backdoor Trigger Inversion Framework. In ICLR 2023
>
> [4] Anti-Backdoor Learning: Training Clean Models on Poisoned Data. In NeurIPS 2021
>
> [5] Delving into Noisy Label Detection with Clean Data. In ICML 2023
>
> [6] Don’t FREAK Out: A Frequency-Inspired Approach to Detecting Backdoor Poisoned Samples in DNN. In CVPR 2023
>
> [7] Spectre: Defending against backdoor attacks using robust statistics. In ICML 2021
>
> [8] Detecting Backdoor Attacks on Deep Neural Networks by Activation Clustering. In AAAI 2019
>
> [9] STRIP: A Defence Against Trojan Attacks on Deep Neural Networks. In ACSAC 2019

---

> ### Comment · Reviewer_mh2m · 2023-11-21
> **It seems quite easy to detect the backdoor.**
>
> Thank you very much for the new results. However, Table 2 shows the backdoor is very easy to detect. This harms the attack strength of the proposed method.
>
> From my perspective, for a backdoor attack to be effective, it should have a reasonably high ASR and not be easily detected.

---

> > ### Author Response · Authors · 2023-11-21
> > **Second Response to Reviewer mh2m**
> >
> > Dear Reviewer mh2m,
> >
> > Thanks for your feedback. For Table 2 in the previous response, we introduce a **new column to denote the adverage ASR** of all detected backdoored models. Additionally, we add a **new row to highlight the relative increage rate**. According to the results in the following table,  we would like to share our thoughts as follows:
> >
> > - **In contrast to the increase in detection accuracy, our enhancement in ASR proves to be more substantial.** The results  reveal a noteworthy **72.4%**  relative increase rate in the average ASR, conspicuously surpassing the relative growth rate  in detection accuracy (**15% and 4.62%**). Considering that the detection accuracy of the baseline is already very high, compared with the significant increase of ASR, the growth of detection accuracy brought by our method is not obvious.
> > -  **We wish to reiterate that the assessment of backdoor attacks encompasses two dimensions: (1)the effectiveness of attacks without defense; (2)the resistance to defenses.**
> >     - **Our paper primarily concentrates on improving the effectiveness of existing backdoor attacks by well-designed selection strategy without changing original triggers.** The comprehensive experiments in the manuscript demonstrate that LPS indeed improves attacks’ effectiveness. LPS can be effortless integrated into any data-poisoning based backdoor attack.
> >     - **The resistance to defenses is likely linked to the strength and stealthiness of triggers.** In our paper, we do not make any assumption of triggers of existing attacks. Significantly, our proposed selection strategy can be combined with more advanced triggers that exhibit greater stealthiness and strength to further enhance resistance to defenses. This is a valuable research direction worthy of exploration in our future work.
> >
> > | Method               | Average ASR (%) | ABS (Detection ACC %) | Unicorn (Detection ACC %) |
> > | -------------------- | --------------- | --------------------- | ------------------------- |
> > | Random               | 32.50           | 53.33                 | 86.67                     |
> > | LPS (Ours)           | 56.03           | 61.33                 | 90.67                     |
> > | **Relative Increase Rate** | **72.4%**       | **15%**               | **4.62%**                 |
> >
> > Thanks again for your valuable feedback.
> >
> > Best regards,
> >
> > Authors

---

### Official Review · Reviewer_n1Ao · 2023-10-31

**Soundness:** 3 good
**Presentation:** 3 good
**Contribution:** 3 good
**Rating:** 6
**Confidence:** 4

**Summary:**

The authors present a sample selection method for data poisoning aimed at enhancing backdoor attacks. A min-max optimization technique is employed to learn a poisoning mask for selecting the appropriate samples.

**Strengths:**

Pros:
- The manuscript is well-organized and easy to follow.
- Although the idea of sample selection for poisoning is conceptually similar to the FUS method, the two approaches diverge in their perspectives. While FUS focuses on local optimization, the proposed method aims for global sample selection.
- The empirical results are robust and substantiate the paper's claims effectively.

**Weaknesses:**

Cons:
- The code for replication is not provided, limiting the paper's reproducibility.
- The significant training loss gap between poisoned and clean samples might make the attack easily detectable by potential victims.
- Ethic statement is missing.

**Questions:**

- Does the threshold "T" vary across different datasets and model architectures?
- Is the proposed approach effective for the combination of CNN-based surrogate models and attention-based target models?
- Could the authors clarify why the method underperforms when the poisoning rate is low?
- For the ablation study, could the authors provide results of LPS\PC?

**Details Of Ethics Concerns:**

Given that this paper focuses on boosting backdoor attacks, an ethical statement should be included.

---

> ### Author Response · Authors · 2023-11-19
> **Response to Reviewer n1Ao (Part 1)**
>
> Dear Reviewer n1Ao,
>
> We sincerely appreciate your precious time and constructive comments, and we are greatly encouraged by your high recognition of **the novel perspective of our approach**, **the robust results**, and **well organization**.
>
> In the following, we would like to answer your concerns separately.
>
> ---
>
> **Q1:** The code for replication is not provided, limiting the paper's reproducibility.
>
> **R1:** Thanks for this suggestion. **Actually, we have provided the code with the paper submission.** You can check the **supplementary material** and unzip the download file to access the code. If the paper is accepted, we will release the code in  github.
>
> ---
>
> **Q2:** The significant training loss gap between poisoned and clean samples might make the attack easily detectable by potential victims.
>
> **R2:** Thanks for this comment. We would like to clarify this concern from the following aspects:
>
> * **Selecting poisoned samples by the attacker and training the target model by the user are two different stages (see Fig 1 in manuscript).** The selection process is inaccessible for the user and defenser. While the poisoned samples are selected based on loss gap, the crafted poisoned dataset are used for training a backdoored model by using standard cross-entropy loss. Therefore, the loss of backdoor samples will decrease as training progresses. **We further empirically show the training loss of poisoned samples selected by random selection and LPS respectively in Table 1 as follows** (The detailed Fig 8 can be found in the revised manuscript). We find that the loss of samples selected by LPS is relatively larger than random selection in the early stage, which aligns with our purpose of identifying hard examples to enhance attack performance. However, the loss of items selected by both methods nearly converges to 0, resulting in the difficulty of detecting the poisoned sample by loss gap.
> - **The performance of backdoor attacks are reflected in two aspects: (1)The effectiveness of attacks without defense.  (2)The resistance to defenses.**  This paper focuses on boosting the attacks’ effectiveness by proposing well-designed selection stragety. The comprehensive experiments in the manuscript demonstrate that LPS indeed improves attacks’ effectiveness. **On the other hand, the resistance to defenses depends more on triggers. However, we do not make any changes to triggers of existing attacks.** Hence, the cababilities of exising attacks to against defenses have not been weakened, which can be demonstrated as follows:
>     - **For existing backdoor defenses, especially for ABL according to training-loss, we achieve similar performance reduction compared to random selection** (see Table 4 in manuscript). It is worth noting that since we have improved the effectiveness of the attacks unser no defense, the attacks are more robust to defenses.
>     - **Additionally, we add comparison on four poisoned sample detection mehtods**, including FREAK[1], Spectral[2], AC[3], Strip[4], and the results are shown in **Table 2** as follows. We find that LPS and random selection shows similar performance on TPR, means **our LPS does not increase the risk of being deteced.**
>     - **Furthermore, we  evaluate on BHN[5], which is a noise detection method**  based on the assumption that the cross-entropy loss of noise sample will be larger than that of normal sample. The results are shown in **Table 3** as follows. We find that the TPR of detecing BadNets is relatively low, indicating that **noise detector is hard to detect poisoned samples selected by our LPS strategy.**
> - In order to improve defense capabilities, we need to focus on designing more stealthy triggers combined with well-designed selection strategies, which remains to be further explored.
>
>
> Table 1: The cross-entropy loss of poisoned samples selected by different strategies on CIFAR-10, where poisoning ratio is set as 0.432% and the target model is ResNet34.
> | Stage| 0| 5| 10| 15| 20| ...  | 75| 80| 85| 90| 95|
> | - | -| -| -| -| -| -| -| -| -| -| -|
> | Random| 3.3336 | 3.3865 | 1.9746 | 1.9746 | 0.3659 | ...  | 0.0011 | 0.0002 | 0.0008 | 0.0008 | 0.0004 |
> | **LPS (Ours)** | 3.8211 | 5.3534 | 4.4994 | 3.1894 | 1.4226 | ...  | 0.0013 | 0.0053 | 0.0009 | 0.0012 | 0.0006 |
>
> Table 2: Results of various backdoor detection method against attacks on CIFAR-10, where poisoning ratio is set as 0.108%.
> || FREAK (TPR%) | Spectral (TPR%) | AC (TPR%) | Strip (TPR%) |
> | -| -| -| -| -|
> | Random (BadNets) | 11.11 | 0    | 0| 0|
> | LPS (BadNets)    | 3.70  | 0    | 0| 0|
> | Random (Blended) | 7.41  | 7.41 | 0| 0|
> | LPS (Blended)    | 9.26  | 7.41 | 0| 0|
>
> Table 3: Detection results of BHN on CIFAR-10, where poisoning ratio is set as 0.216%.
> || BadNets (TPR %) | BadNets (FPR%) |
> | -| -| -|
> | Random         | 11.12| 2.65|
> | **LPS (Ours)** | 11.93| 2.72|
>
> ---

---

> ### Author Response · Authors · 2023-11-19
> **Response to Reviewer n1Ao (Part 2)**
>
> **Q3:** Ethic statement is missing.
>
> **R3:** Thanks for this constructive comment. We claim ethic statement as follows. We will add this section in the revised version.
>
> The adversaries could exploit our proposed LPS strategy to amplify the impact of existing backdoor attack. If LPS strategy are not well defensed, it will poses substantial  threat to modern machine learning systems. This underscores the critical need for the development of proactive defense strategies and detection mechanisms to enhance  the robustness of machine learning systems.
>
> ---
>
> **Q4:** Does the threshold "T" vary across different datasets and model architectures?
>
> **R4:** As detailed in Section 5.1, the threshold 'T' is set as 15 across different datasets and model architectures.
>
> ---
>
> **Q5:** Is the proposed approach effective for the combination of CNN-based surrogate models and attention-based target models?
>
> **R5:** Thank you for this constructive comment. We conduct additional evaluation when the target modes is ViT and the results are shown in **Table 3** as follows. We find that our proposed method still effective in this senario.
>
> Table 3: Attack results on CIFAR-10, where surrogate model is ResNet18 and target model is Vit_b_16, where poisoning ratio is set as 0.216%
> || BadNets (ASR %) | Blended (ASR %) |
> | -------------- | --------------- | --------------- |
> | Random         |39.9|72.47|
> | **LPS (Ours)** |43.42|76.2|
>
> ---
>
> **Q6:** Could the authors clarify why the method underperforms when the poisoning rate is low?
>
> **R6:** Thanks for this constructive comment. We would like to clarify this concern from the following aspects:
>
> - **The performance improvement of selection strategy is influenced  by the initial attack's effectiveness due to the dynamic min-max optimizaiton.** We have briefly explained this phenomeono in the Section 5.2, *i.e., "Specifically, when the poisoning ratio is extremely low (e.g., 1 Img/Cls, 0.054% pratio), although the improvement of our method is not obvious compared with other strategies due to the attack itself being weak, it also shows similar results."*
> - **Specifically, when the poisoning ratio is exceedingly low, the attack itself is weak, it's difficult to inject backdoor with few poisoned samples at the initial minimization step.** Therefore, for subsequent maximization step, it is also hard to learn a reasonable masks given the inherent weakness of the initial attack.
>
> ---
>
> **Q7:** For the ablation study, could the authors provide results of LPS\PC?
>
> **R7:** Thanks for this suggestion. We add  ablation study of LPS\PC and the results are shown in the **Table 4** as follows. **The perclass constraint contributes significantly to LPS strategy.**
>
> Table 4: Ablation studies of LPS’s constraints.
> |Attack| Pratio | LPS（ASR%） | LPS\PC (ASR%) | FUS (ASR%) |  RANDOM (ASR%) |
> | ------- | ------ | ----------- | ------------- | ------------- | ------------- |
> | BadNets | 0.216% | 80.58       | 76.34         | 68.01         | 62.57         |
> | Blended | 0.432% | 87.20       | 86.02         | 79.06         | 75.67         |
>
> ---
>
> Thanks again for your time and attention. We hope the response can address your concerns.
>
> Best regards,
>
> Authors

---

> > ### Comment · Reviewer_n1Ao · 2023-11-22
> > **Reply to authors**
> >
> > Thank the authors for their response. The explanation and results in the response have addressed some of my concerns. However, I am curious about the experimental results in **R5**. Firstly, is the setup of this attack all-to-one or all-to-all? Secondly, there is a significant difference between the results in this table and those in Table 1 of the original paper. Could the author please elaborate on these results and the differences with the resnet18->resnet34 in Table 1? Why is the ASR so much lower on BadNet, but significantly higher on Blended?
> >
> > Additionally, I suggest that the author provide descriptions and explanations for all experimental results, as all the results I see in the response are data-only without explanations.

---

> > > ### Author Response · Authors · 2023-11-22
> > > **Second Response to Reviewer n1Ao**
> > >
> > > Dear Reviewer n1Ao,
> > >
> > > Thanks for your helpful feedback.
> > >
> > > We appreciate your pointing out our mistake in the title of Table 3 in  R5  and apologize for the concern this mistake has caused you.
> > >
> > > In fact, the poisoning rate has been set at **0.432%**. Moreover, BadNets was evaluated in **all-to-all** setting and Blended was evaluated in **all-to-one** setting. We have updated the detailed settings in the below table.
> > >
> > >
> > > Table 3: Attack results on CIFAR-10, where surrogate model is ResNet18 and target model is Vit_b_16.
> > >
> > > |                | BadNets (Pratio=0.432%, all-to-all) | Blended (Pratio=0.432%, all-to-one) |
> > > | -------------- | ----------------------------------- | ----------------------------------- |
> > > | Random         | 39.9                                | 72.47                               |
> > > | **LPS (Ours)** | 43.42                               | 76.2                                |
> > >
> > > Follow your suggestion,  we would like to add more explainations  for experimental results in the response:
> > >
> > > - **Explaination of Table 1 in R1:** Table 1 in R1 shows the training loss of poisoned samples selected by random selection and LPS respectively. We find that the loss of samples selected by LPS is relatively larger than random selection in the early stage, which aligns with our purpose of identifying hard examples to enhance attack performance. However, the loss of items selected by both methods nearly converges to 0, resulting in the difficulty of detecting the poisoned sample by loss gap.
> > > - **Explaination of Table 2 in R1:** Table 2 in R1 shows the comparison on four poisoned sample detection mehtods. Due to the poisoning ratio is low, most detection methods failed. Furthermore, LPS and random selection shows similar performance on TPR, means our LPS does not increase the risk of being deteced.
> > > - **Explaination of Table 3 in R1:** Table 3 in R1 shows the comparison on BHN, which is a SOTA noisy label detection method. Due to poisoned samples comprises different representations compared to normal samples, they are hard to be detected by BHN.
> > > - **Explaination of Table 3 in R5:** Table 3 in R5 shows the attack results when the target model is ViT. Due to attention-based model has different fundamental block compares with convolution-based model, the ASR of ViT  is slightly lower than ResNet34. However, the ASR are still higher than random, indicating the effectiveness of our methods applied on different models.
> > > - **Explaination of Table 4 in R7:** Table 4 in R7 shows the ablation study when the LPS excludes 'perclass' constraint. Without the constraint of averaging selected samples for each class, the selected poisoned samples lack diversity, thus resulting in worse results than LPS, which demonstrates the effectiveness of this constraint.
> > >
> > > Thanks again for your feedback. We hope this clarification can address your concern.
> > >
> > > Best regards,
> > >
> > > Authors

---

> > > > ### Comment · Reviewer_n1Ao · 2023-11-23
> > > > **Reply to authors**
> > > >
> > > > Thank the authors for their efforts in addressing my concerns. I will keep my positive rating for this work.

---

> > > > > ### Author Response · Authors · 2023-11-23
> > > > >
> > > > > Thank you for your thorough review and positive response to our paper.
> > > > >
> > > > > Best regards,
> > > > >
> > > > > Authors

---

> ### Author Response · Authors · 2023-11-21
> **Anticipating Your Feedback**
>
> Dear Reviewer n1Ao,
>
> We would like to express our sincere gratitude for your valuable insights and suggestions on our work.
>
> We have tried our best to address the concerns and queries you raised during the rebuttal process. However, we would greatly appreciate knowing whether our response have effectively resolved your doubts. Your feedback will be instrumental in improving the quality of our work.
>
> We understand the demands of your busy schedule, and we genuinely value your contribution to the improvement of our manuscript. As the end of the discussion period is approaching, we eagerly await your reply before the end.
>
> Thank you once again for your time and attention.
>
> Best regards,
>
> Authors

---

### Official Review · Reviewer_CSPw · 2023-10-31

**Soundness:** 3 good
**Presentation:** 2 fair
**Contribution:** 3 good
**Rating:** 6
**Confidence:** 4

**Summary:**

This paper addresses the inefficiency in existing data-poisoning based backdoor attacks which arbitrarily select samples from a benign dataset to poison, overlooking the varying significance of different samples.

The paper proposes a Learnable Poisoning sample Selection (LPS) strategy. The strategy employs a min-max optimization approach to understand which samples are most crucial for poisoning.

The paper sets up a two-player adversarial game: The inner optimization focuses on maximizing the loss concerning the mask, to pinpoint hard-to-poison samples. The outer optimization aims to minimize the loss with respect to the model's weight, to train the surrogate model.
Through multiple iterations of this adversarial training, the system selects samples that have a higher contribution to the poisoning process.

Comprehensive experiments on established datasets are conducted. Results showcase that the LPS strategy significantly enhances the efficiency and effectiveness of several data-poisoning based backdoor attacks.

**Strengths:**

- The paper proposes a novel approach to select samples for poisoning. The approach is intuitive and effective.
- The paper provides a comprehensive evaluation and solution proof.

**Weaknesses:**

- The problem makes some sense to me, but I am not sure how practical it is. A practical scenario will better motivate the problem.

**Questions:**

1. What is the practical scenario that the proposed boosted attack can be used in?

2. What is the scalability of the proposed attack? For example, the ImageNet has 1000 classes. Is the inner maximization still effective?

3. Some term usages are confusing. For example, under the context of backdoor attack, the term 'm' mask usually refers to the mask of trigger. However, in this paper, the term 'm' mask refers to the mask of sample selection. It would be better to use a different term to avoid confusion. Another example, in page 3, 'K' refers to class numbers. But Section 6 says 'K' is epoch numbers. It would be better to clarify such inconsistency.

---

> ### Author Response · Authors · 2023-11-17
> **Response to Reviewer CSPw**
>
> Dear Reviewer CSPw,
>
> We sincerely appreciate your precious time and constructive comments, and we are greatly encouraged by your high recognition of our proposed **intuitive and effective approach**, **comprehensive evaluation** and **solution proof**.
>
> In the following, we would like to answer your concerns separately.
>
> ---
>
> **Q1:** The problem makes some sense to me, but I am not sure how practical it is. A practical scenario will better motivate the problem. What is the practical scenario that the proposed boosted attack can be used in?
>
> **R1:** Thank you for this insightful comment. This task is very practical in the real world scenario and is clarified as follows:
>
> - **Our proposed LPS strategy can be sued in data-poisoning based backdoor attack.** In this scenario, the attacker can only access and manipulate the training dataset, while the training process is out of control. Specifically, the attacker select partial training samples to be poisoned by adding triggers and changing corresponding labels as target classes.
> - **Data-poisoning based backdoor attack is a practical threat for real world scenarios.** Since modern deep models are often data-hungry, the user may download open-sourced dataset from an unverified source or buy from an untrustworthy third-party data supplier. Considering the data scale, it is difficult to thoroughly check the data quality. In this scenario, the malicious attacker has the chance to manipulate the data to achieve the goal of injecting backdoor into the trained models.
>
> ---
>
> **Q2:** What is the scalability of the proposed attack? For example, the ImageNet has 1000 classes. Is the inner maximization still effective?
>
> **R2:** Thanks for this constructive suggestion. We would like to answer this concern from the following aspects:
>
> - **Theoretically, the number of classes will not affect the inner maximization.** In the inner maximization problem, we decompose it into $K$ independent linear optimization sub-problems, where $K$ is the number of classes. Each linear optimization sub-problem can be solved by a sorting algorithm with lower computational complexity, which is described in sec 4 of the draft. The complexity of each sorting algorithm is $O(n_{k}\log n_{k})$ where $n_{k}$ is the number of sample of label k, and since there are $K$ optimization problems, the overall complexity is $\sum_{k=0}^{K-1}O(n_{k}\log n_{k})\simeq O(N\log(N/K))$ where $N$ is th length of dataset and we assume the dataset exhibits a balanced distribution of samples across various class. Therefore, the number of classes will not affect the optimization in theory.
> - **We empirically evaluate our proposed LPS strategy on full ImageNet-1k with 1000 classes.** Specifically, we choose two samples from each class for poisoning and the poisoning ratio is 0.156% The results are shown in **Table 1** as follows. From the results, we can find that our LPS is still effective  when the number of classes and samples increases to a large size, demonstrating the effectiveness of our approach.
>
> Table 1: Attack results on ImageNet-1k, where surrogate model is ResNet18 and target model is ResNet34.
> |Pratio=0.156%| BadNets (ASR %) | Blended (ASR %) |
> | -------------- | --------------- | --------------- |
> | Random         |81.3|90.21|
> | **LPS (Ours)** |84.02|92.78|
>
> ---
>
> **Q3:** Some term usages are confusing. For example, under the context of backdoor attack, the term 'm' mask usually refers to the mask of trigger. However, in this paper, the term 'm' mask refers to the mask of sample selection. It would be better to use a different term to avoid confusion. Another example, in page 3, 'K' refers to class numbers. But Section 6 says 'K' is epoch numbers. It would be better to clarify such inconsistency.
>
> **R3:** Greatly appreciate your careful inspection. **In Section 6**, we replace the number of epochs 'K' as 'C' to avoid confusion. We will thoroughly proofread the manuscript to correct all notation conflict and update the revised manuscript. All updates are **highlighted with blue in Section 6** of the revised manuscript.
>
> ---
>
> Thanks again for your time and attention. We hope the response can address your concerns.
>
> Best regards,
>
> Authors

---

> > ### Author Response · Authors · 2023-11-22
> > **Looking forward to your reply!**
> >
> > Dear CSPw,
> >
> > We would like to express our gratitude again for your valuable suggestions. As the deadline of ICLR rebuttal period is approaching, we look forward to hearing your feedback on our response. We would be grateful if you could re-evaluate our paper based on our response and new supplementary results.
> >
> > Best regards,
> >
> > Authors

---

> > > ### Comment · Reviewer_CSPw · 2023-11-23
> > >
> > > I appreciate the authors for their efforts in the rebuttal. My concerns are addressed and I keep the positive score.

---

> > > > ### Author Response · Authors · 2023-11-23
> > > >
> > > > Thank you for your thorough review and positive response to our paper.
> > > >
> > > > Best regards,
> > > >
> > > > Authors

---

> ### Author Response · Authors · 2023-11-21
> **Anticipating Your Feedback**
>
> Dear Reviewer CSPw,
>
> We would like to express our sincere gratitude for your valuable insights and suggestions on our work.
>
> We have tried our best to address the concerns and queries you raised during the rebuttal process. However, we would greatly appreciate knowing whether our response have effectively resolved your doubts. Your feedback will be instrumental in improving the quality of our work.
>
> We understand the demands of your busy schedule, and we genuinely value your contribution to the improvement of our manuscript. As the end of the discussion period is approaching, we eagerly await your reply before the end.
>
> Thank you once again for your time and attention.
>
> Best regards,
>
> Authors

---

### Meta-Review · Area_Chair_GUyL · 2023-12-06

**Metareview:**

The paper studies new techniques to enhance backdoor attack. While most prior works randomly select samples to poison, this work presents a min-max formulation that aim to find samples with high loss to poison. It also developed a heuristic alternating optimization algorithm to find a solution. The performance of the approach is evaluated on various data sets and settings.

**Strengths**
- The proposed method is clearly motivated and is easy to follow up.
- In some settings, the method outperforms baselines, showing the potential of it.

**Weaknesses**
- Finding hard samples turns out to be both a strength and weakness of the approach. In some cases, the new attack method is easy to detect.
- In some settings, the improvement of the new attack is minor compared to baselines.
- The attack success rate is low as indicated in Table 4, indicating that the new attack may not be appealing in practice.

**Justification For Why Not Higher Score:**

There are major concerns about the practical performance that authors were unable to address.

**Justification For Why Not Lower Score:**

N/A

---

### Decision · Program_Chairs · 2024-01-16

Reject